# Sex affects transcriptional associations with schizophrenia across the dorsolateral prefrontal cortex, hippocampus, and caudate nucleus

Kynon J. M. Benjamin[1,2,3,8] ✉, Ria Arora[1,4,8], Arthur S. Feltrin [1], Geo Pertea [1], Hunter H. Giles [1,5], Joshua M. Stolz[1], Laura D'Ignazio[1,3], Leonardo Collado-Torres[1,6], Joo Heon Shin [1], William S. Ulrich[1], Thomas M. Hyde[1,2,3], Joel E. Kleinman [1,2], Daniel R. Weinberger [1,2,3,5,7], Apuã C. M. Paquola[1,3] ✉ & Jennifer A. Erwin [1,3,7] ✉

Schizophrenia is a complex neuropsychiatric disorder with sexually dimorphic features, including differential symptomatology, drug responsiveness, and male incidence rate. Prior large-scale transcriptome analyses for sex differences in schizophrenia have focused on the prefrontal cortex. Analyzing BrainSeq Consortium data (caudate nucleus: n = 399, dorsolateral prefrontal cortex: n = 377, and hippocampus: n = 394), we identified 831 unique genes that exhibit sex differences across brain regions, enriched for immune-related pathways. We observed X-chromosome dosage reduction in the hippocampus of male individuals with schizophrenia. Our sex interaction model revealed 148 junctions dysregulated in a sex-specific manner in schizophrenia. Sex-specific schizophrenia analysis identified dozens of differentially expressed genes, notably enriched in immune-related pathways. Finally, our sex-interacting expression quantitative trait loci analysis revealed 704 unique genes, nine associated with schizophrenia risk. These findings emphasize the importance of sex-informed analysis of sexually dimorphic traits, inform personalized therapeutic strategies in schizophrenia, and highlight the need for increased female samples for schizophrenia analyses.

For more than a century, sex differences have been observed in schizophrenia—a complex, chronic neuropsychiatric disorder affecting ~1% of the adult population worldwide. These sex differences include differences in cognitive severity and age of onset; for example, female individuals appearing to be less vulnerable to altered verbal processing deficits[1], and male individuals having an earlier age of disease onset[2,3]. Additionally, prenatal stress may significantly increase the risk of schizophrenia in male offspring as opposed to female offspring[4,5]. To

[1]Lieber Institute for Brain Development, Baltimore, MD, USA. [2]Department of Psychiatry and Behavioral Sciences, Johns Hopkins University School of Medicine, Baltimore, MD, USA. [3]Department of Neurology, Johns Hopkins University School of Medicine, Baltimore, MD, USA. [4]Department of Biology, Johns Hopkins University Krieger School of Arts & Sciences, Baltimore, MD, USA. [5]Department of Genetic Medicine, Johns Hopkins University School of Medicine, Baltimore, MD, USA. [6]Center for Computational Biology, Johns Hopkins University, Baltimore, MD, USA. [7]Department of Neuroscience, Johns Hopkins University School of Medicine, Baltimore, MD, USA. [8]These authors contributed equally: Kynon J. M. Benjamin, Ria Arora. ✉e-mail: KynonJade.Benjamin@libd.org; Apua.Paquola@libd.org; Jennifer.Erwin@libd.org

date, only two large-scale RNA-sequencing studies have examined sex differences in schizophrenia, and both focus exclusively on one brain region—the prefrontal cortex[6,7]. Furthermore, the Genotype-Tissue Expression (GTEx) analysis of sex differences across 45 tissues found fewer than 100 differentially expressed genes (DEGs) in 13 of 14 brain regions. This small number of identified DEGs might be attributed to relatively limited sample size (114 to 209 individuals)[8].

Leveraging schizophrenia genome-wide association studies (GWAS)[9–11], recent large-scale studies have used statistical association between genotype and expression (expression quantitative trait loci [eQTL]) to identify genomic features (gene, transcript, exon, and exon-exon junctions) underlying schizophrenia risk[12–16]. However, these studies have not explored potential sex-interacting eQTL (si-eQTL). Furthermore, the GTEx study across 44 tissues found only four si-eQTL genes with a nominally significant false discovery rate (FDR) below 0.25 in only two of the 13 brain regions examined[8]. Other recent si-eQTL studies involving whole blood[17,18] and lymphoblastoid cell lines[19] have found fewer than 25 si-eQTL. As eQTL have tissue specificity[20], the field of neuropsychiatric genetics needs a sizable and comprehensive analysis of si-eQTL in the human brain.

Here, we leverage the BrainSeq Consortium RNA-sequencing and genotypes datasets to identify genes associated with sex, with sex-specific expression in schizophrenia, and with si-eQTL using a total of 1170 samples across 504 individuals (Table 1) for the caudate nucleus ($n = 399$), dorsolateral prefrontal cortex (DLPFC; $n = 377$), and hippocampus ($n = 394$). Our work increases the number of annotated sex-biased features, examines sex-chromosome dosage, identifies sex-specific schizophrenia features, provides annotations of si-eQTL in the human DLPFC and hippocampus, and increases si-eQTL annotations for the caudate nucleus. Altogether, these results provide insights into sex differences, highlighting the importance of sex-informed analysis of sexually dimorphic traits and informing personalized therapeutic strategies in schizophrenia.

## Results

### Sex-specific expression across the caudate nucleus, DLPFC, and hippocampus

We first explored sex differences in the brains of the 480 unique individuals (caudate nucleus [$n = 393$], DLPFC [$n = 359$], and hippocampus [$n = 375$]) by performing differential expression of sex after adjusting for diagnosis, age, ancestry (SNP PCs 1–3), RNA quality, and hidden variances (Eq. 1 and Table 3) using the BrainSeq Consortium dataset[12,13]. We observed 831 unique DEGs (FDR < 0.05; Fig. 1A) between the sexes across the caudate nucleus ($n = 689$ DEGs [279 upregulated in females; 410 upregulated in males]), DLPFC ($n = 256$ [99 upregulated in females; 157 upregulated in males]), and hippocampus ($n = 147$ [64 upregulated in females; 83 upregulated in males]). Of these 831 unique DEGs, the sex chromosomes showed the most significant sex-biased expression (Data S1). Interestingly, most sex-associated DEGs for the caudate nucleus and DLPFC were autosomal (Table S2). When we expanded our analysis to the isoform level, we identified an

additional 859 unique genes associated with a differentially expressed transcript, exon, or exon-exon junction (Fig. S4). Furthermore, we observed a similar pattern of majority autosomal genes; however, the most significant differentially expressed features in this context were located on sex chromosomes (Table S2 and Data S1).

To evaluate the function and association of the sexually dimorphic gene expression to heritable complex traits, we performed gene set enrichment analysis and MAGMA enrichment analysis on DEGs separately per brain region. For MAGMA gene set enrichment, we did not observe any significant associations (Data S2). This may be partly explained by the underrepresentation of allosomes in GWAS studies[21]. With gene set enrichment analysis, we observed significant enrichment (GSEA, $q < 0.05$) of several ontology and disease terms for each brain region (Fig. 1B and Data S3). As expected, we found enrichment for dosage compensation associated with female-biased (upregulated in females) DEGs across all brain regions. Additionally, we found significant enrichment of fibroid related disease terms (i.e., uterine fibroids, tubulointerstitial fibrosis, and fibroid tumor) associated with female-biased DEGs across all brain regions. The vast majority of enrichment terms (1477 of 1515 ontology and disease terms) were associated with female-bias DEGs across brain regions. The small fraction of enrichment terms associated with male-biased DEGs included histone modification, androgen receptor signaling, and autistic behavior (Data S3). In contrast, female-biased DEGs were primarily enriched for immune-related pathways (i.e., neutrophil migration, macrophage activation, humoral immune response, complement activation, and astrocyte development) across the brain (Data S3). We also found these enrichment patterns replicated in co-expression network modules[22] associated with sex (Data S4). Even so, female- and male-constructed network modules were well preserved ($Z > 10$; Data S5), similar to reports by others[23]. Altogether, these results imply that networks shared between female and male individuals are well preserved, while significant sex-specific modules are highly enriched for immune-related pathways.

### Autosomes influence sex differences in the brain

With so many autosomal sex-biased DEGs, we asked to what degree sex-biased autosomal genes contribute to sex differences in the brain. We separated allosomal and autosomal DEGs and performed principal component analysis to assess explained variance of these DEGs across the brain. While the first principal component of all allosome DEGs explained ~97% variance across brain regions, the autosomal DEGs also showed significant association with sex (Fig. S6). Interestingly, we found that explained variance drastically increased with only the 10 most significant autosomal DEGs across brain regions (90%, 87%, and 88% for the caudate nucleus, DLPFC, and hippocampus; Fig. 1C and Fig. S6). As it appeared that a small subset of these autosomal DEGs explained a large proportion of expression variances between the sexes, we formally tested this using dynamic recursive feature elimination (dRFE)[24]. To this end, we applied random forest classification using 10-fold, sex-stratified, cross-validation with dRFE and found a

**Table 1 | A sample breakdown of eQTL analysis for individuals (age > 13) postmortem caudate nucleus, DLPFC, and hippocampus from the BrainSeq Consortium, separated by sex**

| Brain Region | Sex | Sample Size | Diagnosis | Ancestry | Age (mean ± sd) | RIN (mean ± sd) |
|---|---|---|---|---|---|---|
| Caudate Nucleus | F | 126 | 76CTL/50SZ | 79AA/47EA | 50.2 ± 16.9 | 7.8 ± 0.9 |
| | M | 273 | 169CTL/104SZ | 127AA/146EA | 48.6 ± 15.7 | 7.9 ± 0.8 |
| DLPFC | F | 121 | 73CTL/48SZ | 75AA/46EA | 48.4 ± 17.1 | 7.4 ± 1.0 |
| | M | 256 | 156CTL/100SZ | 129AA/127EA | 44.6 ± 16.1 | 7.8 ± 0.9 |
| Hippocampus | F | 126 | 79CTL/47SZ | 82AA/44EA | 47.9 ± 16.7 | 7.64 ± 1.1 |
| | M | 268 | 182CTL/86SZ | 131AA/137EA | 44.4 ± 16.2 | 7.7 ± 1.0 |

*F* female, *M* male, *CTL* neurotypical control, *SZ* schizophrenia, *AA* African American, *EA* European American, *RIN* RNA integrity number.

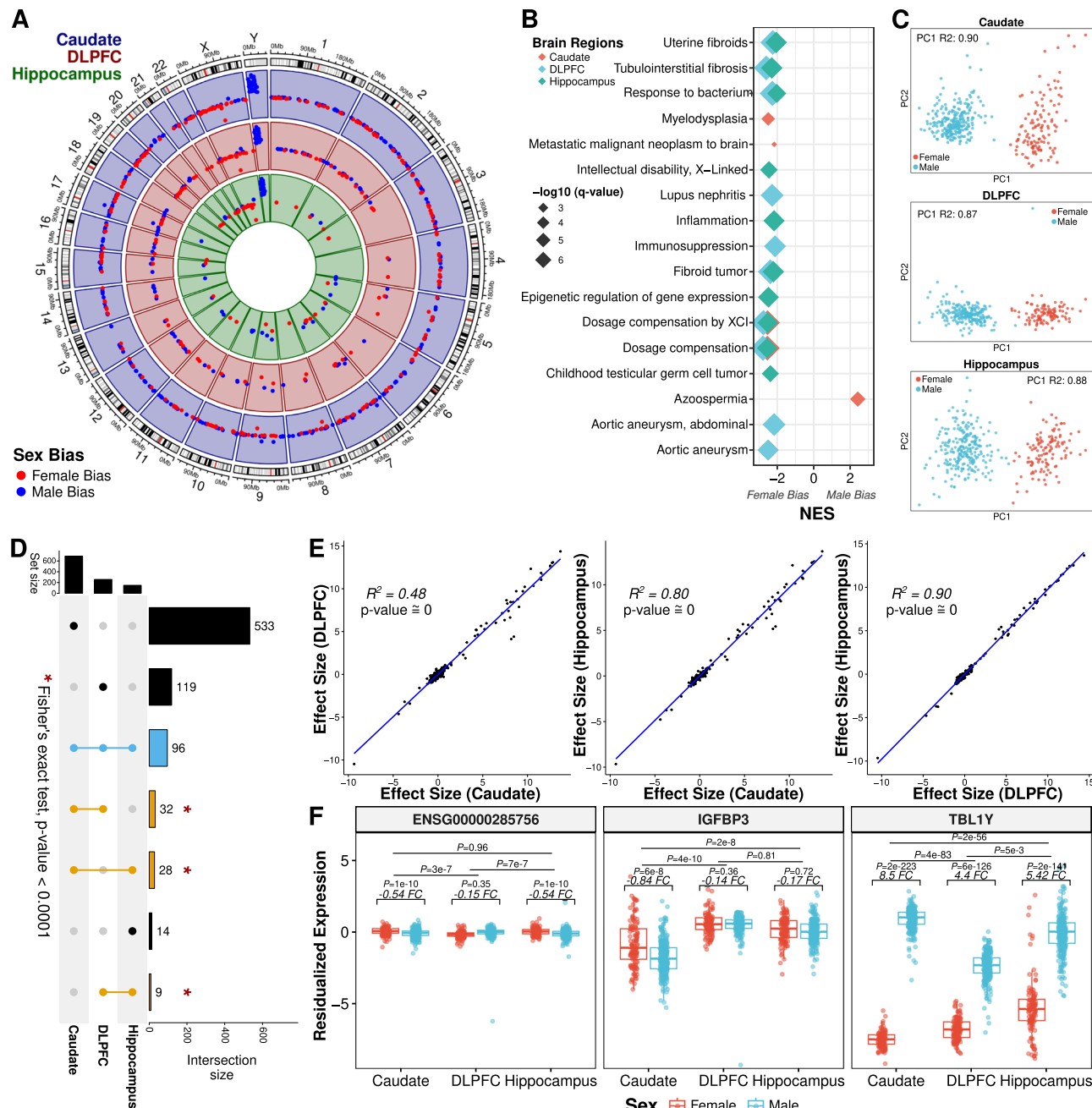

**Fig. 1 | Sex-biased expression across the caudate nucleus, DLPFC, and hippocampus.** **A** Circos plot showing significant differentially expressed genes (DEGs) for the caudate nucleus (blue; *n* = 393; 121 female and 272 male), DLPFC (dorsolateral prefrontal cortex; red; *n* = 359; 114 female and 245 male), and hippocampus (green; *n* = 375; 121 female and 254 male) across all chromosomes. Female bias (upregulated in female individuals) in red, and male bias (upregulated in male individuals) in blue. **B** Gene set enrichment analysis (GSEA) of sex differential expression analysis across brain regions, highlighting the top ten most significant terms upregulated in females (female bias) or males (male bias). NES normalized enrichment score, XCI X-chromosome inactivation. **C** Scatterplots of the estimated proportion of expression variance explained by sex within the 100 most significant autosomal DEGs (i.e., adjusted *p* value) for the caudate nucleus (DEGs, *n* = 60), DLPFC (DEGs, *n* = 25), and hippocampus (DEGs, *n* = 42). Plots are annotated with parametric correlation (*R*²) from linear regression for PC1. **D** UpSet plot showing overlap of DEGs across the caudate nucleus, DLPFC, and hippocampus. Blue is shared across the caudate nucleus, DLPFC, and hippocampus; orange, shared between two brain regions; and black, unique to a specific brain region. * Indicating significant enrichment (caudate nucleus vs DLPFC: *p* < 2.3e−141; caudate nucleus vs hippocampus: *p* < 1.1e−181; and DLPFC vs hippocampus: *p* < 9.3e−190) using two-sided, Fisher's exact test. **E** Scatterplot of effect size (logFC) for all genes tested showing concordant positive directionality with significant two-sided, Spearman correlation (*R*²) of all genes. A fitted trend line is presented in blue as the mean values ± standard deviation. **F** Example box plots of genes showing an interaction between sex and brain region (caudate nucleus: *n* = 393 [121 female and 272 male]; DLPFC: *n* = 359 [114 female and 245 male]; and hippocampus: *n* = 375 [121 female and 254 male]). FC = fold change log2 (male/female). Female individuals in red and male individuals in blue. Adjusted *p* value (*P*) annotation using dream[67] (default of Satterthwaite approximation) generated statistics annotation. Box plots show the median and first and third quartiles, and whiskers extend to 1.5× the interquartile range.

median of 110 genes (77, 32.5, and 221 for caudate nucleus, DLPFC, and hippocampus respectively) with perfect test score accuracy for sex classification (Table S3 and Fig. S7). Interestingly, the prediction accuracy seemed to be driven by one pseudogene (*RPS10P3* [ribosomal protein S10 pseudogene 3]) shared across brain regions (Fig. S8). Additionally, *RPS10P3*—located on chromosome 9—has previously been reported to be associated with five different traits[25–29], including sex-interacting cleft lip[29] and lateral ventricle temporal horn volume in psychosis[28]. These results indicate that a small subset of autosomal genes significantly contributes to sexually dimorphic gene expression in the brain.

## Brain region interaction with sex
To understand the regional specificity of sex DEGs, we compared DEGs from each brain region. We observed a significant enrichment of shared DEGs across the three brain regions (Fisher's exact test, $p < 0.05$; Fig. 1D) with the majority (60 DEGs, 62.5%) on sex chromosomes. For replication analysis, we compared the DEGs with previous sex differences analysis in the brain[6,30–32] and found greater than 62% of DEGs were significantly differentially expressed in all brain regions except for the GTEx cerebellum and anterior cingulate cortex (Fig. S9) with a concordant direction of effect between BrainSeq Consortium and GTEx brain regions (Fisher's exact test, $p < 0.01$). For a more in-depth comparison, we examined the sex differences found using the CMC DLPFC[6]. We also discovered a large number of DEGs on sex chromosomes (39 of 51 [76.5%] and 41 of 54 [75.9%] for the NIMH HBCC and MSSM-Penn-Pitt cohorts, respectively) similar to our BrainSeq Consortium analysis. Additionally, we observed significant pairwise enrichment of these CMC DEGs with our BrainSeq Consortium DEGs across brain regions (Fisher's exact test, $p < 0.01$; Fig. S10). Altogether, this suggests that X- and Y-linked genes drive brain-wide sex expression differences and autosomal genes drive brain region-specific differences. Additionally, autosomal DEGs were less likely to replicate in different datasets.

Interestingly, we found that all genes, regardless of significant association with sex, showed a significant positive correlation for the direction of effect between pairwise comparisons of the three brain regions (Spearman, rho $> 0.69$, $p < 1.4e-108$; Fig. 1E). At significant levels (DEGs, adjusted $p < 0.05$), these pairwise correlations dramatically increased (Spearman, rho $> 0.99$, $p < 4e-104$; Fig. S11A) with significant concordant direction of effect (RRHO; Fig. S12). Expanded analysis of transcripts, exons, and exon-exon junctions displayed a similar pattern with all shared, differentially expressed (DE) features (adjusted $p < 0.05$) having significant concordant direction and significant positive correlation among brain regions (Spearman, rho $> 0.97$, $p \cong 0$; Fig. S11B–D). Moreover, at significant levels (adjusted $p < 0.05$), all directions agreed between the CMC DLPFC and the BrainSeq Consortium brain regions with a significant positive correlation (Spearman; rho $> 0.97$ for all pairwise comparisons; $p < 1.1e-44$; Fig. S13). In summary, the direction of change for sexually dimorphic genes is generally shared across multiple brain regions and independent datasets.

We next evaluated the degree of sex bias among brain regions formally with an interaction model for sex and brain region. Here, we found extensive interactions (Fig. 1F), particularly in the caudate nucleus as compared to the DLPFC (adjusted $p < 0.05$, DEGs = 528) and the caudate nucleus as compared to the hippocampus (adjusted $p < 0.05$, DEGs = 71). In contrast, we found only five genes (*ZNF736P9Y*, *ENSG00000285756*, *TUBBP1*, *TBL1Y*, and *ENSG00000285679*) with region-specific expression for sex between the DLPFC and hippocampus. When we expanded our analysis to the isoform level, we identified more than double unique DEGs (1303 [775 isoform only], 198 [127 isoform only], and 23 [18 isoform only], for caudate nucleus vs DLPFC, caudate nucleus vs hippocampus, and DLPFC vs hippocampus, respectively; Fig. S14, Table S4, and Data S6).

To understand the functional significance of these brain region-specific transcriptional changes, we applied GSEA and MAGMA enrichment for each pairwise comparison. While we did not find enrichment for any brain or non-brain traits (MAGMA; Data S2), we did observe significant enrichment (GSEA, $q < 0.05$) of several ontology terms for all pairwise comparisons (Fig. S15 and Data S7). Interestingly, we observed terms associated with myelination (i.e., myelination, axon ensheathment, and ensheathment of neurons) for comparisons between the DLPFC and the other two brain regions. For brain region-specific transcriptional sex differences between the caudate nucleus and hippocampus, we also observed enrichment for cognition, neurotransmission, and regulation of synaptic plasticity. For transcriptional sex differences between the caudate nucleus and DLPFC, we observed additional terms associated with gene silencing. For transcriptional sex differences between the DLPFC and hippocampus, we observed additional terms associated with receptor signaling. Altogether, this analysis highlights the importance of brain region-specific transcriptional sex differences, which are significantly enriched for neurotransmission and myelination.

## XCI and dosage compensation in the brain
As 70% of the brain-wide, sexually dimorphic genes are located on sex chromosomes, we next evaluated the dosage of X-linked genes compared to autosomes. In order to equalize the dosage of X-linked genes between XX females and XY males, female mammals epigenetically silence one X chromosome in a process called XCI (X-chromosome inactivation). XCI is a chromosome-wide process where the majority of X-linked genes are nearly completely silenced, and a minority of X-linked genes either escape X inaction or show variable X inactivation. When we examined the DEGs by brain region for X-linked gene dosage, we found the majority of DEGs were enriched for XCI escape genes (Fig. S16, Fisher's exact test, Bonferroni < 0.01; Fig. 2A), reflecting dosage compensation for the majority of X-linked genes subject to XCI as seen in previous studies[33,34].

Across all three brain regions, we found XCI escape genes were significantly enriched within the female- and male-biased DEGs (Fisher's exact test, Bonferroni < 0.01; Fig. 2A). Moreover, all male-biased escaping XCI genes were located on the PAR (pseudoautosomal regions) of both X and Y chromosomes (*AKAP17A*, *ASMTL*, *ASMTL-AS1*, *CD99*, *CD99P1*, *DHRSX*, *GTPBP6*, *IL3RA*, *LINC00106*, *PLCXD1*, *PPP2R3B*, and *ZBED1*; Data S8). This finding aligns with previous reports showing a male-biased enrichment of escaping XCI genes on PAR1[35,36]. Additionally, we found the most male-biased XCI-annotated genes in the caudate nucleus ($n = 16$, 9, and 10 DEGs for caudate nucleus, DLPFC, and hippocampus, respectively; Fig. S17), which were mostly annotated as escape XCI genes (Data S8 and Fig. S17). In contrast, we only found enrichment of variable XCI genes in the caudate nucleus (Fisher's exact test, Bonferroni < 0.01). Altogether, XCI escaping genes demonstrated higher expression in female individuals across brain regions, suggesting sex differences shared across brain regions are associated with well-documented XCI escaping genes for females.

Next, we evaluated differences in chromosome-wide dosage by comparing the relative X chromosome expression (RXE) to autosomes (Fig. 2B and Fig. S18). Interestingly, we observed a significant decrease of RXE in male individuals only in the DLPFC (Mann-Whitney U, $p = 0.047$), demonstrating region-specific dosage compensation. We also observed a similar trend of decreased RXE in the DLPFC from the CMC MSSM-Penn-Pitt cohort (Mann-Whitney U, $p = 0.07$; Fig. S19A) but not the GTEx frontal cortex (Fig. S20). Even so, the large RXE variation across the 13 GTEx brain regions demonstrated region-specific dosage compensation (Fig. S20).

As we found differences in the DLPFC between sexes, we next asked if this might be due to individuals with schizophrenia. Interestingly, we found decreased RXE in the hippocampus of male patients (Mann-Whitney U, $p = 0.004$; Fig. 2C) but not in the caudate nucleus,

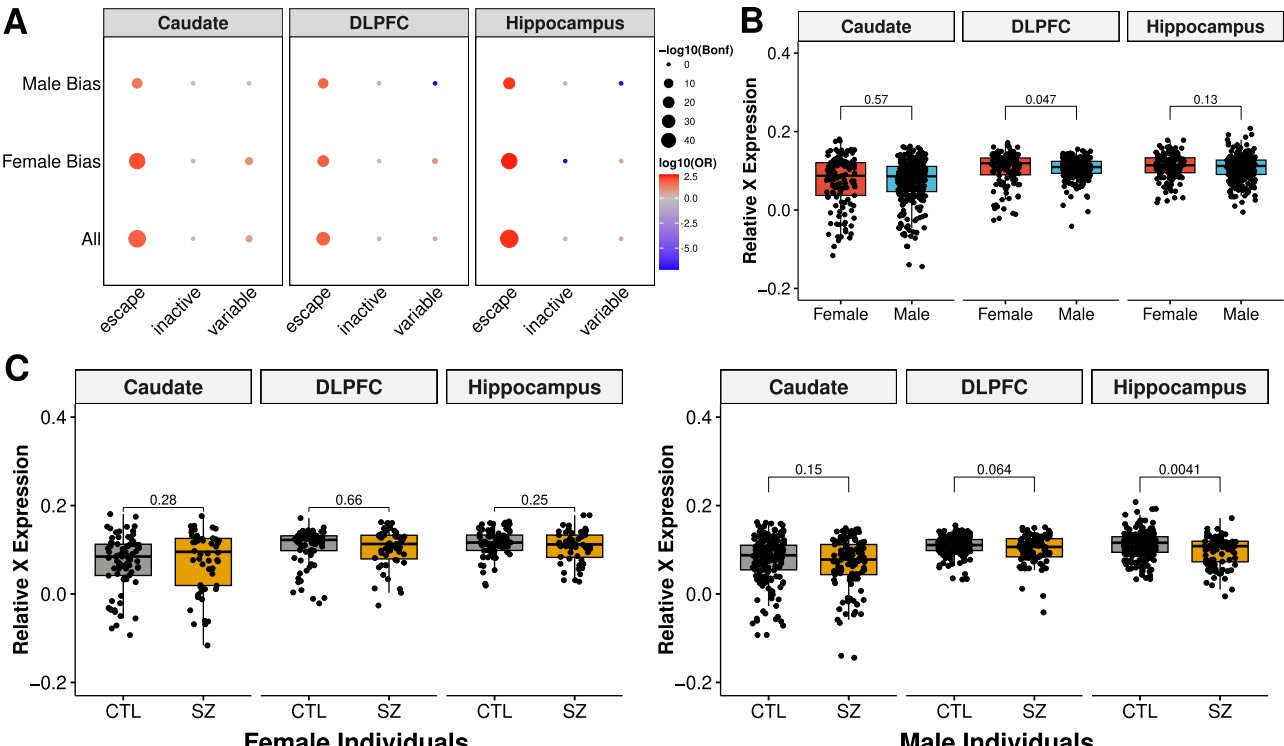

**Fig. 2 | X-linked gene expression and dosage compensation observed across the caudate nucleus, DLPFC, and hippocampus. A** Enrichment of significant sex-biased genes relative to genes known to escape X-chromosome inactivation (XCI) across the caudate nucleus (n = 393; 121 female and 272 male), DLPFC (dorsolateral prefrontal cortex; n = 359; 114 female and 245 male), and hippocampus (n = 375; 121 female and 254 male). Dot plot of enrichment (two-sided, Fisher's exact test) of differentially expressed genes (DEGs) for XCI genes by brain region separated by male bias (upregulated in male individuals), female bias (upregulated in female individuals), and all DEGs. Size of dots denotes -log10 of Bonferroni corrected p values. Color relates to log10 odds ratio (OR) with depletion in blue and enrichment in red. **B** Box plot showing relative X-chromosome expression (RXE) comparison between female (red) and male (blue) individuals with two-sided, Mann-Whitney U two-tailed, p values annotated across the caudate nucleus (n = 393; 121 female and 272 male), DLPFC (dorsolateral prefrontal cortex; n = 359; 114 female and 245 male), and hippocampus (n = 375; 121 female and 254 male). **C** Box plot showing significant differences between neurotypical control (CTL; gray) and schizophrenic (SZ; gold) individuals in the caudate nucleus (n = 393; 121 female and 272 male), DLPFC (n = 359; 113 female and 245 male), and hippocampus (n = 375; 121 female and 254 male) for female (left) and male (right) individuals with annotation of two-sided, Mann-Whitney U p values. All box plots show the median and first and third quartiles, and whiskers extend to 1.5× the interquartile range.

DLPFC, or CMC DLPFC (Fig. S19B). This significant decrease in RXE in the hippocampus of male patients with schizophrenia was driven by the reduction of expression from inactive XCI genes (Fig. S21). However, there was no significant interaction between sex and diagnosis status for any brain region for RXE. These results demonstrate slight differences between X-chromosome dosage in the hippocampus of individuals with schizophrenia.

### Interaction of schizophrenia and sex in the brain

After investigating sex differences in the brain without consideration of diagnosis in 480 unique individuals (caudate nucleus [n = 393], DLPFC [n = 359], and hippocampus [n = 375]), we next identified statistically significant differentially expressed features (adjusted p < 0.05) with respect to sex differences and diagnosis through an interaction model. No genes, transcripts, or exons were significant by this interaction model, similar to a previous study[6]. While overall replication with CMC DLPFC was limited, we found significant correlation of nominally significant (p < 0.05) transcriptional signatures between DLPFC and CMC DLPFC, NIMH HBCC cohort (π1 = 0.51; Spearman, ρ = 0.60, p < 0.01; Fig. S22 and Table S5). In contrast, on the junction level, 148 junctions demonstrated a significant (adjusted p < 0.05; Fig. S23A and Data S9) interaction between sex and diagnosis across the caudate nucleus (nine unannotated junctions; Fig. S23B), DLPFC (89 unannotated junctions; Fig. S23B), and hippocampus (47 unannotated junctions and three junctions associated with *DDX11L1* [DEAD/H-Box Helicase 11 Like 1]; Fig. S23B).

We also examined differential expression for schizophrenic female and male individuals separately across the caudate nucleus, DLPFC, and hippocampus using RRHO analysis to increase our power of detecting transcriptional changes. Here, we found schizophrenia-related transcriptional signatures, while concordant direction of effect varied dramatically depending on sex and brain region. Specifically, female schizophrenia transcriptional signatures showed the strongest pattern of sharing between the caudate nucleus and the DLPFC (Fig. 3A), while males showed the strongest pattern of sharing between DLPFC and hippocampus (Fig. 3B). These patterns are similar to those found in ours and others' previous schizophrenia analyses adjusted for sex[12,13]. Altogether, sex-adjusted schizophrenia analysis largely reflects male transcriptional changes likely due to larger male sample sizes.

Next, we examined female and male transcriptional changes for schizophrenia within individual brain regions. Here, we found the strongest shared signature within the caudate nucleus with very little observable overlap for the DLPFC and hippocampus (Fig. 3C). Furthermore, we examined DEGs (adjusted p < 0.05) and found a large overlap for the caudate nucleus but little overlap between females and males for the DLPFC and hippocampus (Fig. 3D). For the DLPFC and hippocampus, the limited observable overlap is apparent in the opposing enrichment patterns between DLPFC and hippocampus in females (Fig. 3A) and males (Fig. 3B); while female transcriptional signatures associated with genes upregulated in schizophrenia, male transcriptional signatures associated with genes downregulated.

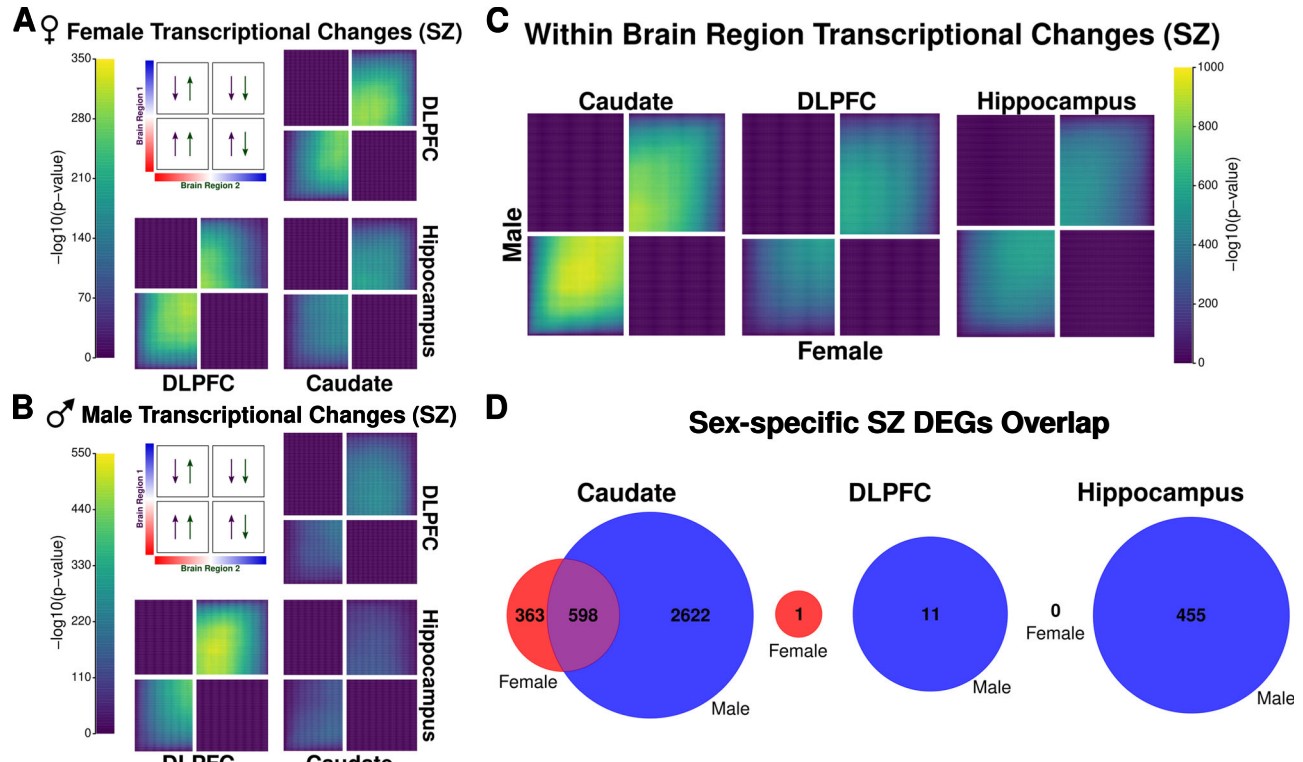

**Fig. 3 | Transcriptional changes for schizophrenia shared between sexes within brain regions.** RRHO (rank-rank hypergeometric overlap) maps comparing schizophrenia transcriptional changes for all genes between brain region pairs stratified by direction of effect in (**A**) females and (**B**) males. The panel presents the overlapping relationship between two brain regions. The color bars represent the degree of significance [-log10(*p* value)] of overlap between two brain regions. Arrows show the direction of effect for schizophrenia (upregulated or downregulated in schizophrenia) by brain region. RRHO uses two-sided, hypergeometric testing. **C** RRHO map comparing female and male schizophrenia transcriptional

changes within brain regions for all genes. The color bar represents the degree of significance [-log10(*p* value)] of the overlap between the sexes. RRHO uses two-sided, hypergeometric testing. **D** Venn diagram showing overlap within brain regions for sex-stratified schizophrenia differentially expressed genes (DEGs; female in red and male in blue; FDR < 0.05). Female-specific schizophrenia DEGs in red, male-specific schizophrenia DEGs in blue, and schizophrenia DEGs shared between female- and male-specific schizophrenia analyses in purple. SZ schizophrenia.

Similar trends occurred for transcripts, exons, and junctions (Data S10 and Fig. S24).

## Sex-specific schizophrenia expression in the brain

To further examine female- and male-specific schizophrenia DEGs, we applied a more stringent filter to exclude: (1) genes shared between female-only and male-only schizophrenia analyses and (2) genes detected as significantly different (Mann-Whitney U, *p* < 0.05) with residualized expression for opposite sex-only individuals. For female-specific schizophrenia analysis, the single DLPFC DEG and 169 DEGs from the caudate nucleus were filtered out because they showed significant residualized expression for male individuals with schizophrenia (Mann-Whitney U, *p* < 0.05) resulting in a total of 194 female-specific schizophrenia DEGs in the caudate nucleus (Table S6 and Data S10). For male-specific schizophrenia analysis, we found 1130, 5, and 149 DEGs for the caudate nucleus, DLPFC, and hippocampus, respectively. Due to the small number of DLPFC-identified DEGs, we found no DEGs shared across brain regions for male-specific schizophrenia analysis (Table S7). We found two genes (*EDN3* and *PLD4*) shared between the caudate nucleus and DLPFC and 23 genes (Table S7) shared between the caudate nucleus and hippocampus.

We hypothesized that the smaller sample size for female individuals might explain why we identified few if any female-specific schizophrenia DEGs within the BrainSeq Consortium dataset. To test this hypothesis, we performed 1000 random samplings of the male individuals at female sample sizes (*n* = 121, *n* = 114, and *n* = 121, for the

caudate nucleus, DLPFC, and hippocampus, respectively) and calculated DEGs for each brain region and permutation. On average, we identified a drastically smaller number of schizophrenia DEGs (median male-only schizophrenia DEGs of 347, 0, and 1 for the caudate nucleus, DLPFC, and hippocampus, respectively; Fig. S28) in our subsampled male samples that showed no significant difference (permutation *p* = 0.30, 0.80, and 0.69 for the caudate nucleus, DLPFC, and hippocampus, respectively) between the number of schizophrenia DEGs identified from the female-only analysis. Altogether, the smaller female sample size, at least partially, explains the lack of identification of female-specific schizophrenia DEGs within the BrainSeq Consortium datasets.

Given the role of estrogen in the treatment of schizophrenia[37], we next examined sex hormone expression in the context of schizophrenia in the brain. Here, we did not observe any sex-specific or sex-specific schizophrenia DEGs for sex hormone expression (*AR* [androgen receptor], *ESR1* [estrogen receptor 1], *ESR2* [estrogen receptor 2], and *PGR* [progesterone receptor]) across the brain. As sex hormone levels vary with age, we next examined any potential interaction between sex hormones and age. For female and male individuals, we found no significant interactions of diagnosis status and age for the caudate nucleus and DLPFC. For the hippocampus, we found a nominally significant upregulation of *ESR2* (linear regression, *p* = 0.016 [FDR = 0.098]; Fig. S25) in female individuals with schizophrenia compared with neurotypical controls and *PGR* in male individuals with schizophrenia compared with neurotypical controls as a function of age (linear regression, *p* = 0.013 [FDR = 0.16]; Fig. S26). These results

suggest that sex hormones expression may potentially interact with age and diagnosis status in the hippocampus.

We next examined functional and MAGMA enrichment of these sex-specific, schizophrenia transcriptional changes. For MAGMA gene set enrichment, we found significant enrichment of schizophrenia and neutrophils—a white blood cell type—traits for male-specific DEGs upregulated in individuals with schizophrenia for the caudate nucleus and hippocampus (Data S2). We also found significant enrichment of the schizophrenia trait for female-specific DEGs upregulated in individuals with schizophrenia for the caudate nucleus (Data S2). In contrast, we found significant enrichment of basophil and eosinophils for male-specific DEGs downregulated in individuals with schizophrenia for the DLPFC (Data S2). With these immune-related cell type enrichment, we were not surprised to observed significant depletion (GSEA, $q < 0.05$) of immune-related pathways for both female- and male-only schizophrenia analysis (Fig. S27A,B and Data S11). Interestingly, we found the hippocampus showed the greatest degree of similarity between female- and male-only schizophrenia functional enrichment analysis (best-match average > 64%; Fig. S27C). In contrast, the caudate nucleus showed the lowest degree of similarity between sex-specific schizophrenia analysis with biological processes and molecule function showing 42% and 38% similarity in GO terms, respectively. This seemingly contradictory analysis highlights the potential impact of the larger number of sex-specific schizophrenia DEGs identified in the caudate nucleus (Fig. 3D). In addition to this functional analysis, we also investigated gene co-expression networks between female and male individuals by diagnosis. While we observed complete preservation of modules for neurotypical control individuals across brain regions, we found one module significantly not preserved ($Z < 10$) for the DLPFC (94 genes [46% protein coding]; Data S5). While we did not find any GO terms significantly enriched, this module included terms related to DNA binding (i.e., *HDDC3*, *MAZ*, *PCBP1*, *TOP3B*, and zinc finger proteins) and transposable and repetitive elements (i.e., *LRRC24* and *TIGD7*).

We next compared our results with the recent meta-analysis for sex-specific schizophrenia DEGs in the prefrontal cortex[7]. Of the 46 male-specific DEGs identified by Qin et al., we found a total of three overlapping genes: one gene overlapping (*PARD3*) with the caudate nucleus stringent female-specific DEGs and two overlapping genes (*USE1* and *ABCG2*) with the hippocampus stringent male-specific DEGs, which all shared direction of effect. When we compared the full set of female and male schizophrenia DEGs across brain regions, we found an additional three overlapping genes (*CD99*, *GABARAPL1*, and *LIN7B*) shared with the caudate nucleus. Of these three only *GABARAPL1* had a discordant direction of effect.

## Sex-dependent eQTL in the brain

We asked whether genetic regulation of expressed features would manifest differently in females compared to males for the caudate nucleus, DLPFC, and hippocampus. We tested for statistical interaction between genotype and sex in the brain by applying multivariate adaptive shrinkage (mash) modeling in the 504 individuals (age > 13) for the caudate nucleus ($n = 399$), DLPFC ($n = 377$), and hippocampus ($n = 394$). We identified hundreds of sex-interacting variants (si-eQTL) across brain regions for gene-, transcript-, exon-, and junction-level analysis (Table 2 and Data S12). For example, we found 703, 545, and 546 gene-level si-eQTL (local false sign rate [lfsr] < 0.05) for the caudate nucleus, DLPFC, and hippocampus, respectively, accounting for 704 unique genes with si-eQTL (eGenes; Table 2 and Data S12) driven by the caudate nucleus. Only 21 (3.0%) of these eGenes (si-eQTL associated with unique genes) were located on the X chromosome; the majority of eGenes were located on autosomes, similar to sex-specific expression analysis. We found this proportion aligns with the ratio of autosomes to allosomes and shows no significant shift in distribution

**Table 2 | Summary of sex-interacting eQTL (lfsr <0.05) across brain regions for genes, transcripts, exons, and exon-exon junctions associated with all si-eQTL**

| Brain Regions | | Caudate Nucleus | DLPFC | Hippocampus |
|---|---|---|---|---|
| Gene | eQTL | 3274 | 2464 | 2465 |
| | eFeature | 703 | 545 | 546 |
| | eGenes | 703 | 545 | 546 |
| Transcript | eQTL | 10,186 | 8032 | 8207 |
| | eFeature | 1700 | 1383 | 1394 |
| | eGenes | 1577 | 1286 | 1296 |
| Exon | eQTL | 10,439 | 7873 | 7824 |
| | eFeature | 1737 | 1426 | 1420 |
| | eGenes | 801 | 665 | 662 |
| Junction | eQTL | 1740 | 1150 | 1145 |
| | eFeature | 328 | 228 | 228 |
| | eGenes | 7 | 5 | 5 |

eQTL: number of variant-feature pairs for each feature type: genes, transcripts, exons, and exon-exon junctions. eFeature: number of unique features that have eQTL associations. eGene: number of eQTL associations with unique genes. lfsr: local false sign rate[45].

between eGenes and genes tested (Kolmogorov-Smirnov test, $p = 0.89$).

To understand the regional specificity of these si-eQTL, we examined the proportion of si-eQTL detected across brain regions. Here, we found the majority (544 [77%]) of eGenes were shared across brain regions (Fig. 4A), which was also observed on the isoform level (i.e., transcripts, exons, and junctions; Fig. S29). Remarkably, all of the shared si-eQTL showed concordant directionality. Furthermore, the DLPFC and hippocampus showed nearly identical si-eQTL effect sizes (Fig. 4B), which was confirmed with the high level of replication across brain regions ($\pi 1 > 0.996$; Fig. S30). The few brain region-specific si-eQTL showed small but significant sexual dimorphic genetic regulation of expression (Fig. S31). Unsurprisingly, the exon- and junction-level sharing showed a smaller proportion of shared si-eQTL across brain regions at an effect size within a factor of 0.99 (Fig. S32), suggesting alternative isoform usage drives differences in si-eQTL effect size across brain regions.

To evaluate the functional relevance underlying si-eQTL in the caudate nucleus, DLPFC, and hippocampus, we performed functional gene term enrichment analysis on the eGenes for each brain region. We observed significant enrichment (hypergeometric, adjusted $p < 0.05$) across brain regions (Data S13), including enrichment for neurogenesis and cellular localization (Fig. 4C). Notably, we found that these enriched GO terms showed high semantic similarity[38] (best-match average, 62–100%) across brain regions (Fig. S33). While we did not find these si-eQTL associated eGenes significant enrichment for sex-specific DEGs (Fisher's exact test, $p > 0.10$), we did found significant enriched (Fisher's exact test, FDR < 0.05) for neurological disorders including schizophrenia[12–14,39], autism spectrum disorder[39], and Alzheimer's disease[40] (Fig. S34).

When we compared our si-eQTL with previous work in whole blood and in lymphoblastoid cell lines, we found no overlap with the 19 si-eQTL identified in whole blood[17,18] and two genes (*ATG4C* and *CA2*) of the 21 si-eQTL identified in lymphoblastoid cell lines[19] and also present in the caudate nucleus si-eQTL (Data S14). We next compared our results with the four si-eQTL ($q < 0.25$) identified in GTEx brain regions (amygdala and nucleus accumbens basal ganglia)[8] and found no overlaps. When we expanded to the 369 si-eQTL ($q < 0.25$) from all 43 GTEx tissues[8], we found two overlapping genes encoding noncoding RNAs (*ENSG00000270605* and *ENSG00000272977*) between the caudate

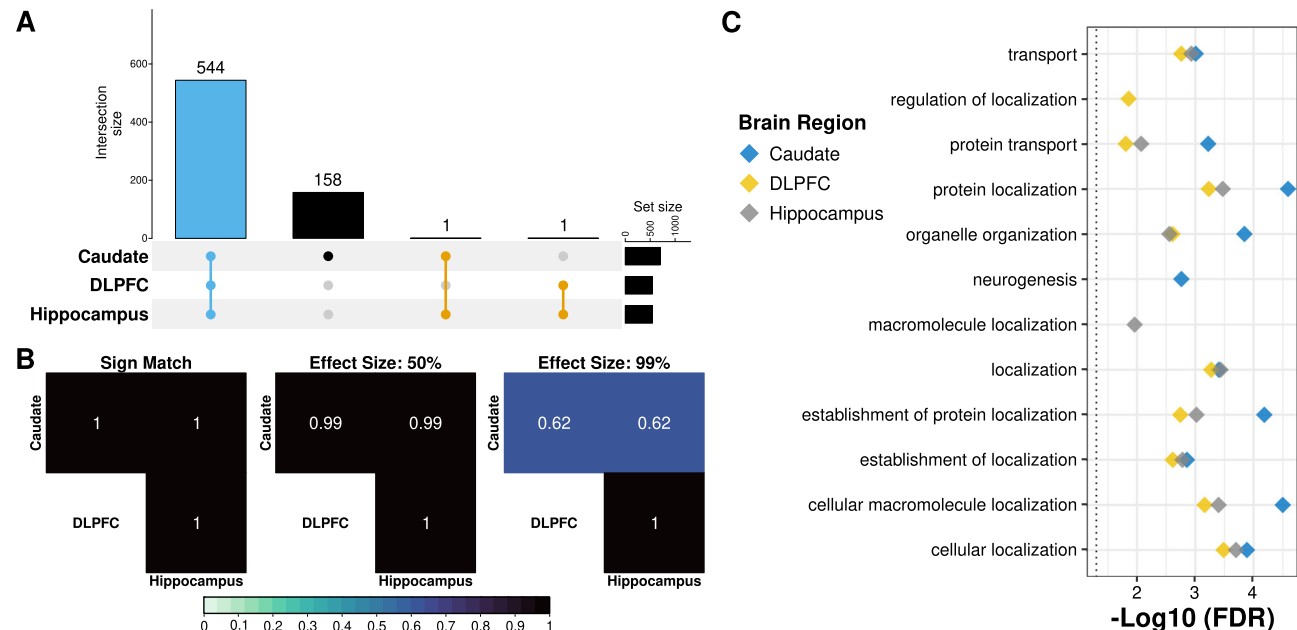

**Fig. 4 | Sex-interacting eQTL (si-eQTL) are shared across brain regions. A** UpSet plot displaying overlap across brain regions for si-eQTL (local false sign rate [lfsr] < 0.05). Blue is shared across the caudate nucleus (n = 399; 126 female and 273 male), DLPFC (dorsolateral prefrontal cortex; n = 377; 121 female and 256 male), and hippocampus (n = 394; 126 female and 268 male); orange, shared between two brain regions; and black, unique to a specific brain region. **B** Heatmap of the proportion of gene level si-eQTL sharing with sign match (left), within a factor of 0.5 effect size (middle), and within a factor 0.99 effect size (right). **C** Functional enrichment plot of the ten most significant gene ontology-terms (biological processes) of eGenes for the caudate nucleus (blue), DLPFC (yellow), and hippocampus (gray).

nucleus and suprapubic skin and spleen GTEx tissues, respectively (Data S14). Relatively low replication rate with GTEx brain regions can, in part, be attributed to low sample sizes in the GTEx dataset[41].

We next set out to determine if any of these si-eQTL had causal associations with schizophrenia risk (PGC3 GWAS p < 5e−8)[11]. To formally identify variants associated with schizophrenia risk, we performed colocalization analysis on fine-mapped gene level si-eQTL across brain regions (Data S15). We identified nine unique genes across the three brain regions with significant colocalization (regional colocalization probability [RCP] > 0.5; Data S16): one genes (ENSG00000287222; Fig. S35) in the caudate nucleus; seven genes in the DLPFC (ACE, ELAC2, FURIN, MMD, ZSCAN29, LINC00320, and ENSG00000289128; Fig. S36); and five genes in the hippocampus (ACE, ELAC2, MMD, STRC, and ENSG00000287222; Fig. S37). Interestingly, the brain regions with the largest detection of schizophrenia risk genes were the DLPFC and hippocampus; three of the nine colocalized genes were shared between these two brain regions. Additionally, the colocalized gene identified in the caudate nucleus (ENSG00000287222) was also significant in the hippocampus. Furthermore, we also identified this shared gene as a sex-specific schizophrenia DEG (male-specific, adjusted p = 0.026) in the caudate nucleus. While we did not observe any tissue-specific overlap between the other eight colocalized genes and sex-specific schizophrenia DEGs—potentially due to low number of identified DEGs—we did observe one additional colocalized gene (FURIN; male-specific, adjusted p = 0.023) as a sex-specific schizophrenia DEG in the caudate nucleus. Interestingly, we found ACE to be downregulated in schizophrenia for both sexes in the caudate nucleus. Altogether, this correlates with the high level of sharing of si-eQTL across brain regions, suggesting that while sex-interacting colocalized genes are highly recurrent across the brain, sex-specific dysregulation of these si-eQTL may be brain-region specific.

## Discussion
Sex has been associated with differential gene expression in the brain and sex-specific effects in neuropsychiatric disorders like schizophrenia. Here, we aimed to take a holistic exploratory analysis approach to sex differences for schizophrenia in the caudate nucleus, DLPFC, and hippocampus. We identified numerous genetic features (genes, transcripts, exons, and exon-exon junctions) that (1) are associated with sex, (2) demonstrate sex-specific expression in schizophrenia, and (3) have expression that interacts with genotypes and sex as si-eQTL. Furthermore, we identified nine genes showing sex-variable association with schizophrenia risk[11] (PGC3 GWAS p < 5e-8) with colocalization and some overlap with sex-specific schizophrenia DEGs. Additionally, we have demonstrated slight differences between X-chromosome dosage in the hippocampus of male individuals with schizophrenia—and further shown that the degree of dosage compensation varies across the brain. To the best of our knowledge, this is the largest multi-brain region analysis for sex differences in schizophrenia.

In this study, we found 831 unique genes with sex-associated differential expression in the caudate nucleus, DLFPC, and hippocampus. While genes on sex chromosomes have the largest sex-specific effects, we also found that autosomal genes significantly influenced sex differences. Our results support previous findings of sex differences, including differences that are brain-region specific[32], are primarily located on autosomes[6,30,31], and have exhibited shared direction of effect driven by allosomal DEGs[32]. Notably, the autosomal pseudogene RPS10P3 emerged as a key driver of sex prediction across the brain. Located in a gene-poor region, RPS10P3 is flanked by enhancers and has prior links to diverse traits[25–29], including sex-interacting cleft lip[29] and psychosis-related lateral ventricle temporal horn volume[28]. A larger number of samples in our cohort allowed us to perform machine learning and identify many more genes than previous analyses conducted across multiple regions, contributing to our knowledge on sex-specific genomic features and sex differences in the brain.

To determine if sex differences observed across brain regions were related to XCI, we further examined DEGs located on the X chromosome. Here, our analysis aligned with previous work showing an enrichment of genes known to escape XCI[33,34], including male-bias

enrichment for PAR genes[35,36]. Additionally, we found that across the brain (i.e., GTEx and BrainSeq Consortium) X-chromosome dosage showed brain region-specific dosage compensation levels.

The second aim of the study was to identify any sex-specific schizophrenia genomic features across multiple brain regions. As one of the most consistently implicated brain regions for the pathophysiology of schizophrenia, the DLPFC has been the primary focus of postmortem analysis for schizophrenia[42,43]. However, we have recently shown the importance of analyzing multiple brain regions, specifically for the identification of additional therapeutic targets[12]. As such, we aimed to identify (1) which brain region showed the most sex-specific schizophrenia-associated transcriptional changes and (2) the degree of sharing of these schizophrenia-associated transcriptional changes across brain regions.

While our interaction model found only junctions (148 exon-exon junctions) significantly dysregulated with a sex-specific interaction across brain regions, comparing female to male expression in schizophrenia allowed us to identify many more sex-specific schizophrenia genomic features than previous studies—even after applying an additional, more stringent filter to our sex-specific schizophrenia DEG results. This was unsurprising as we have previously found the caudate nucleus has substantially more schizophrenia DEGs compared to the DLPFC and hippocampus[12]. While we expect the majority of the sex-specific schizophrenia DEGs in the caudate nucleus to be associated with antipsychotic treatment[12,44], our results also provide a starting point to examine differential antipsychotic effects by sex for schizophrenia.

As we compared multiple brain regions for sex-specific schizophrenia analysis, we also found that sex-specific schizophrenia differentially expressed features were highly brain-region specific. This also was not unexpected as schizophrenia DEGs, irrespective of sex, are also highly brain-region specific[12,13,15]. However, the smaller number of identified DE features for transcripts, exons, and junctions was surprising. This might be due to our study being underpowered, as we found twice as many schizophrenia differentially expressed features in male individuals as compared to females.

Given the role of estrogen in the treatment of schizophrenia, we also examined sex hormone-related expression (*AR*, *ESR1*, *ESR2*, and *PGR*) in the brain. Using expression data, we were unable to identify any significant associations after correcting for multiple testing. We found only a nominally significant interaction with diagnosis and age of *ESR2* in female individuals and *PGR* in male individuals in the hippocampus. This may be due to: (1) conventional methods for hormone measurements using serum, or (2) limited sample size for interaction models. Altogether, these findings suggest a need for increased samples from female individuals to further advance our understanding of potential sex differences for schizophrenia across multiple brain regions.

The third aim of our study was to identify si-eQTL across the brain. We annotated hundreds of si-eQTL associated with 704 eGenes across the caudate nucleus, DLPFC, and hippocampus using mash modeling[45]. With mash modeling, we were able to increase our power to detect si-eQTL. With this study, we have provided annotations of si-eQTL in the DLPFC and hippocampus and found that these sex-interacting eGenes were enriched for neurogenesis as well as DEGs from neurological studies (i.e., schizophrenia, autism spectrum disorder, and Alzheimer's). Our results demonstrate the power of tissue-specific si-eQTL and its potential for identifying genes with sexually dimorphic expression for neurological disorder.

In addition to annotating hundreds of si-eQTL, we have also provided annotations of sex-interacting genes with causal variants associated with schizophrenia risk. One limitation of this analysis is our small female sample size. Even so, these results highlight the importance of examining genes associated with schizophrenia risk for

potential differences in expression for population covariates (e.g., sex, age, and genetic ancestry).

In summary, we have provided a comprehensive genetic and transcriptional analysis of sex differences in schizophrenia. We have increased the number of annotated features exhibiting sex bias in the brain adding to our current understanding of sex differences in the brain, identified sex-specific schizophrenia genes with indications for additional therapeutic targets, and provided annotations of si-eQTL for the DLPFC and hippocampus. These results have the potential to direct therapeutics and strategies that can address sex-biased responses in the treatment of schizophrenia. Additionally, these results highlight the need for more female samples for schizophrenia analyses.

## Methods

The research described herein complies with all relevant ethical regulations. All specimens used in this study were obtained with informed consent from the next of kin under protocols No. 12–24 from the Department of Health and Mental Hygiene for the Office of the Chief Medical Examiner for the State of Maryland and No. 20111080 for the Western Institutional Review Board for the Offices of: (1) the Chief Medical Examiner for Kalamazoo Michigan, (2) University of North Dakota in Grand Forks North Dakota, and (3) Santa Clara County California. Details of case selection, curation, diagnosis, and anatomical localization and dissection can be found in previous publications from our research group[12,13].

### BrainSeq Consortium RNA-sequencing data processing
We surveyed data from the BrainSeq Consortium[12,13] for caudate nucleus, DLPFC, and hippocampus, specifically: phenotype information, FASTQ files, region-specific covariates, and single nucleotide polymorphism (SNP) array genotypes.

We re-mapped RNA-sequencing reads to the hg38/GRCh38 human reference genome (GENCODE release 41, GRCh38.p13) with the splice-aware aligner HISAT2[46] (version 2.2.1). Following alignment, we collected quality control and alignment metrics for each sample using RNA-SeQC[47] (version 2.4.2).

We performed quantification of major genomic features (genes, transcripts, exons, and exon-exon junctions) for each sample separately, as follows:
- We generated gene and exon read counts using featureCounts[48] (version 2.0.3) with default parameters for paired-end, reverse-stranded read counting.
- We estimated transcript expression (i.e., counts and transcripts per million [TPM]) with kallisto[49] (version 0.46.2) with default parameters for reverse-stranded reads.
- We extracted exon-exon junction coverage data for all spliced alignments found in the alignment files produced by HISAT2 using RegTools[50] (version 0.5.3). Using this reference-free method, we were able to include splicing patterns detected from the aligned files.

Following this quality control and quantitation, we packaged these data (i.e., counts, gene annotation, and quality control metrics) into RangedSummarizedExperiment R objects[51] using R code adapted from the SPEAQeasy RNA-seq processing pipeline[52].

### BrainSeq Consortium imputation and genotype processing
We imputed genotypes as previously described[12]. Briefly, we first converted genotype positions from hg19 to hg38 with liftOver[53]. The Trans-Omics for Precision Medicine (TOPMed) imputation server[54–56] was used for imputation of genotypes filtered for high quality (removing low-quality and rare variants) using the genotype data phased with the Haplotype Reference Consortium (HRC) reference

panels (https://mathgen.stats.ox.ac.uk/impute/1000GP_Phase3.html). Genotypes were phased per chromosome using Eagle[57] (version 2.4). We performed quality control with the McCarthy Tools (https://www.well.ox.ac.uk/~wrayner/tools/HRC-1000G-check-bim-v4.3.0.zip): specifically, we removed variants and samples with minor allele frequency (MAF) less than 0.01, missing call frequencies greater than 0.1, and Hardy-Weinberg equilibrium below a $p$ value of 1e-10 using PLINK 2.0[58–60] (version 2.00a3LM). This resulted in 11,474,007 common variants.

For population stratification of samples, we performed multidimensional scaling with PLINK version 1.9[58–60] on linkage disequilibrium (LD)-independent variants. The first component separated samples by ancestry as reported by the medical examiner's offices.

## Sample selection

We selected samples from the caudate nucleus, DLPFC, and hippocampus based on four inclusion criteria: (1) used RiboZero RNA-sequencing library preparation, (2) features an age greater than 13 years, (3) has a self-reported ethnicity of either African American or White American, and (4) has TOPMed imputed genotypes available. This resulted in a total of 1,170 samples from 504 unique individuals across the three brain regions for eQTL analysis. For expression-based analysis, we excluded individuals with ages less than 17 years, resulting in a total of 1127 samples from 480 unique individuals across the three brain regions.

## Subject details

Of all 1,170 samples used in the eQTL portion of this study, 399 were from the caudate nucleus, 377 from the DLPFC, and 394 from the hippocampus. Out of the 1,170 samples, 126, 121, and 126 were female, and 273, 256, and 268 were male from the caudate nucleus, DLPFC, and hippocampus, respectively (Table 1). For the 1,127 samples used in the expression analyses of this study, 393, 359, and 375 samples were located in the caudate nucleus, DLPFC, and hippocampus, respectively (Table 3). More information can be found in Tables 1 and 3. Individual-level, de-identified sample information is provided in Source Data.

## Match gender phenotype to sex chromosomes

To match gender phenotype with sex chromosomes, we applied the sex imputation function (--check-sex) from PLINK. This compares sex assignments in the input dataset with those imputed from X-chromosome inbreeding coefficients. We used a Jupyter Notebook (version 6.0.2) with the R kernel to compare reported gender with genotype-imputed sex (F estimates). Here, we found all gender phenotypes matched sex chromosomes with F estimates for females below 0.22 and males above 0.9 (Fig. S1).

## Quality control and covariate exploration for sex

Observed expression measurements can be affected by biological and technical factors. To evaluate potential confounders for expression or sex, we first correlated technical and RNA quality variables (i.e., RIN, mitochondria mapping rate, overall mapping rate, total gene assignment, mean 3' bias, etc.) and removed highly correlated variables (Pearson, $r > 0.95$; Fig. S2) present in at least one brain region. To examine potential confounders, we next correlated

these remaining variables and biological variables (i.e., diagnosis, age at death, global genetic similarity, status of antipsychotics at time of death, etc.) with gene expression as a function of sex (Fig. S3A). For model covariates, we used variables that had a significant correlation (Bonferroni corrected $p < 0.05$) with gene expression for either sex in at least one brain region. To account for possible hidden effects on gene expression not captured by the above covariates, we also applied surrogate variable analysis[61,62]. When we regressed out biological, technical, and hidden effects, we found this reduced all spurious correlations (Fig. S3B).

## Expression normalization

For expression normalization, we constructed edgeR[63,64] objects in R (version 4.2) for each brain region by using raw counts and sample phenotype information. Next, we filtered out low expression counts using filterByExpr from edgeR (version 3.40.2), which keeps features above a minimum of 10 count-per-million (CPM) in 70% of the smallest group sample size (i.e., female individuals). Following this, we normalized library size using trimmed mean of M-values (TMM) before applying voom normalization[65] using limma[66] (version 3.54.1) on four different linear models that examine: (1) sex (Eqs. 1), (2) interaction of brain region and sex (Eqs. 2), (3) interaction of sex and diagnosis (Eqs. 3), (4) diagnosis subset by sex (Eq. 4). Example covariates for these linear models are diagnosis, age, brain region, genetic similarity (SNP PCs [principle components] 1–3), RNA quality (RIN, mitochondria mapping rate, gene assignment rate, genome mapping rate, rRNA mapping rate, and mean 3' bias). For sex, interaction of sex and diagnosis, and diagnosis by sex analyses, we also corrected for any hidden variance via surrogate variable analysis.

$$
\begin{aligned}
E(Y) = {} & \beta_0 + \beta_1 Sex + \beta_2 Diagnosis + \beta_3 Age + \beta_4 MitoRate + \beta_5 rRNArate \\
& + \beta_6 TotalAssignedGenes + \beta_7 RIN + \beta_8 OverallMappingRate \\
& + \beta_9 Mean3Bias + \sum\nolimits_{i=1}^{3} \eta_i snpPC_i + \sum\nolimits_{j=1}^{k} \gamma_j SV_j
\end{aligned}
\tag{1}
$$

$$
\begin{aligned}
E(Y) = {} & \beta_0 + \beta_1 Sex * \beta_2 Region + \beta_3 Diagnosis + \beta_4 Age + \beta_5 MitoRate \\
& + \beta_6 rRNArate + \beta_7 TotalAssignedGenes + \beta_8 RIN \\
& + \beta_9 OverallMappingRate + \beta_{10} Mean3Bias + \sum\nolimits_{i=1}^{3} \eta_i snpPC_i
\end{aligned}
\tag{2}
$$

$$
\begin{aligned}
E(Y) = {} & \beta_0 + \beta_1 Sex * \beta_2 Diagnosis + \beta_3 Age + \beta_4 MitoRate + \beta_5 rRNArate \\
& + \beta_6 TotalAssignedGenes + \beta_7 RIN + \beta_8 OverallMappingRate \\
& + \beta_9 Mean3Bias + \sum\nolimits_{i=1}^{3} \eta_i snpPC_i + \sum\nolimits_{j=1}^{k} \gamma_j SV_j
\end{aligned}
\tag{3}
$$

$$
\begin{aligned}
E(Y) = {} & \beta_0 + \beta_1 Diagnosis + \beta_2 Age + \beta_3 MitoRate + \beta_4 rRNArate \\
& + \beta_5 TotalAssignedGenes + \beta_6 RIN + \beta_7 OverallMappingRate \\
& + \beta_8 Mean3Bias + \sum\nolimits_{i=1}^{3} \eta_i snpPC_i + \sum\nolimits_{j=1}^{k} \gamma_j SV_j
\end{aligned}
\tag{4}
$$

**Table 3 | A sample breakdown of expression analysis for adult (age > 17) postmortem caudate nucleus, DLPFC, and hippocampus from the BrainSeq Consortium**

| Brain Region | Sample Size | Diagnosis | Sex | Race | Age (mean ± sd) | RIN (mean ± sd) |
|---|---|---|---|---|---|---|
| Caudate Nucleus | 393 | 239CTL/154SZ | 121F/272M | 205AA/188EA | 49.6 ± 15.6 | 7.9 ± 0.9 |
| DLPFC | 359 | 211CTL/148SZ | 114F/245M | 200AA/159EA | 47.4 ± 15.4 | 7.7 ± 0.9 |
| Hippocampus | 375 | 242CTL/133SZ | 121F/254M | 207AA/168EA | 47.0 ± 15.3 | 7.6 ± 1.0 |

*CTL* neurotypical control, *SZ* schizophrenia, *F* female, *M* male, *AA* African American, *EA* European American, *RIN* RNA integrity number.

As there was significant overlap of individuals among the three brain regions examined, we used *dream* (differential expression for repeated measures) from variancePartition[67] (version 1.28.7) to correct for the random effect of duplicate individuals across brain regions to assess potential significant interactions between brain region and sex (Eq. (2)). As such, we applied voom via dream framework with `voomWithDreamWeights`.

## Expression residualization

For residualized expression, we used voom-normalized expression and null models to regress out covariates as previously described[12]. After regressing out covariates, we applied a z-score transformation. Null models were created without variable(s) of interest to examine: (1) sex (Eq. (4)), (2) interaction of brain region and sex (Eq. (5)), (3) interaction of sex and diagnosis (Eq. (6)), (4) diagnosis (Eq. (6)).

$$
\begin{aligned}
E(Y) = \beta_0 &+ \beta_1 Diagnosis + \beta_2 Age + \beta_3 MitoRate + \beta_4 rRNArate \\
&+ \beta_5 TotalAssignedGenes + \beta_6 RIN + \beta_7 OverallMappingRate \\
&+ \beta_8 Mean3Bias + \sum_{i=1}^{3} \eta_i snpPC_i
\end{aligned}
$$

(5)

$$
\begin{aligned}
E(Y) = \beta_0 &+ \beta_1 Age + \beta_2 MitoRate + \beta_3 rRNArate \\
&+ \beta_4 TotalAssignedGenes + \beta_5 RIN + \beta_6 OverallMappingRate \\
&+ \beta_7 Mean3Bias + \sum_{i=1}^{3} \eta_i snpPC_i + \sum_{j=1}^{k} \gamma_j SV_j
\end{aligned}
$$

(6)

## Differential expression analysis

Following voom normalization, we fit four linear models (Eqs. (1)–(4)) to examine: (1) sex, (2) interaction of sex and brain region, (3) interaction of sex and diagnosis, and (4) diagnosis subset by sex. With our fitted model, we identified differentially expressed features using the *eBayes*[68] function from limma. Dream enabled us, in one step, to complete linear model fitting and differential expression calculation for interaction of sex and brain region.

## Weighted correlation network analysis (WGCNA) analysis

We performed a signed network WGCNA[22] (version 1.72) analysis using residualized expression (Eq. (4)) to generate the co-expression network using all genes in a single block by brain region. First, we filtered genes and outlier individuals with the WGCNA function `goodSamplesGenes`. To remove any outlier individuals whose expression substantially deviated from the norm, we also filtered individuals with Z-normalized expression greater than 2.5. After evaluating power and network connectivity for each brain region, we selected a soft-thresholding power of eight for network constructions. We constructed networks using `bicor` correlation and set `deepSplit` to two for the caudate nucleus and hippocampus and three for the DLPFC. Additionally, we set `mergeCutHeight` to 0.15 and `minModuleSize` set to 50 for all brain regions and gene networks. We made the co-expression networks using Pearson correlation values with 381, 349, 364 samples and 23,488, 23,039, and 22,990 genes for the caudate nucleus, DLPFC, and hippocampus, respectively. Significant associations with sex were determined using a linear model and Pearson correlation between binary sex and module eigengenes.

For module preservation analysis[23] between sex (female versus male), we constructed networks across brain regions for all samples, control-only samples, schizophrenia-only samples using the following parameters: (1) soft-thresholding power of 15 and (2) `nPermutations` set to 100. Similar to our signed network construction, we first filtered genes and outlier individuals with `goodSamplesGenes`. As the DLPFC failed to achieve a scale-free topology across networks (i.e., all samples, control-only samples, or schizophrenia-only samples), we also

removed outlier individuals after visual inspection of the sample dendrogram generated with `flashClust` (Fig. S5). This resulted in the removal of one female neurotypical control (17 years; DLPFC) individual and two individuals with schizophrenia: one female (80 years; DLPFC) and one male (66 years; DLPFC). For each brain region, we used the male-generated networks as the reference group. For the neurotypical control analysis, we generated networks with 240 (71 female; 169 male), 210 (66 female; 146 male), and 243 (74 female; 169 male) individuals for the caudate nucleus, DLPFC, and hippocampus, respectively. For the schizophrenia analysis, we generated networks with 153 (50 female; 103 male), 146 (48 female; 98 male), and 132 (47 female; 85 male) individuals for the caudate nucleus, DLPFC, and hippocampus, respectively. For the combined analysis (control and schizophrenia), we generated networks with 393 (121 female; 272 male), 356 (112 female; 244 male), and 375 (121 female; 254 male) individuals for the caudate nucleus, DLPFC, and hippocampus, respectively. We generated all networks with a total of 26,881, 26,627, and 26,727 genes for the caudate nucleus, DLPFC, and hippocampus, respectively. We considered a module to not be preserved if the Z-summary score was less than or equal to ten.

## Multi-marker analysis of genomic annotation (MAGMA) enrichment analysis

For gene set enrichment analysis comparisons of DEGs and multiple GWAS summary statistics, we applied MAGMA[69] (version 1.10) on GWAS SNP *p* values with European reference data downloaded from MAGMA. As the GWAS summary statistics were on hg19, we mapped our DEG gene locations from hg38 to hg19 using the GENCODE v41 GRCh37 lifted annotation file. Initially, we generated a SNP annotation file with the annotate flag (`--annotate`). Following this, we performed gene analysis on SNP *p* values using this SNP-level gene annotation file with the following: (1) PLINK input files (`--bfile`), (2) gene model set to SNP-wise mean, and (3) expanded gene boundaries (35 kb upstream and 10 kb downstream). We used default settings for all other parameters. Once completed, we performed gene-set enrichment analysis in MAGMA (`--gene-results`) using default parameters. We analyzed sex-specific DEGs by direction of effect (upregulated in females or males) across the caudate nucleus, DLPFC, and hippocampus with 12 traits (seven neuropsychiatric; Table S1). We executed this MAGMA pipeline using snakemake[70] (version 6.4.1).

## Random forest dynamic recursive feature elimination

For autosomal sex prediction, we used dRFEtools[24] (version 0.1.17) in Python (version 3.7) to apply dynamic recursive feature elimination with random forest classification[71]. We set the elimination rate to 10% and set 0.30 as the fraction of samples used for lowess smoothing. To reduce overfitting, we generated 10 sex-stratified folds for cross-validation with the `StratifiedKFold` function from scikit-learn (version 1.0.2)[72]. Model performance was measured using normalized mutual information, accuracy, and area under the receiver operating characteristic (ROC) curve with out-of-bag samples.

## X-chromosome inactivation (XCI) enrichment analysis

For XCI enrichment analysis, we downloaded the XCI status annotation from ref. 33. We accessed the enrichment of sex bias for XCI status using Fisher's exact test with the known XCI categories, including 631 genes defined as escape ($n = 99$), variable escape ($n = 101$), or inactive ($n = 431$). We corrected for multiple testing with the Bonferroni procedure.

## Dosage compensation

Relative X expression (RXE) was determined as previously described[73] with slight modifications; specifically, we used transcripts per million (TPM). We generated TPM using the mean of the read insert size for effective length (Eq. 7). We extracted the average fragment length as estimated by kallisto per brain region. We dropped any genes with

effective lengths less than or equal to one. Following TPM calculation, we performed a log2 transformation (Eq. (8)). Next, we filtered low-expressing genes present in at least 20% of samples. To compute RXE, we calculated the differences in the mean chromosome-wide log2 TPM expression with X-chromosome log2 TPM expression (Eq. (9)).

$$\text{Effective Length} = \text{Length} - [\text{Mean Fragment Length}] + 1 \quad (7)$$

$$TPM = 1e6 * \frac{Count/EffectiveLength}{\sum(Count/EffectiveLength)} \quad (8)$$

$$RXE = \log 2(\text{mean TPM of X-chromosome genes}) \\ - \log 2(\text{mean TPM of all autosomal genes}) \quad (9)$$

### Sex-specific differential expression analysis for schizophrenia

To determine more stringent sex-specific differential expression features among the three brain regions using diagnosis subset by sex, we applied additional selection criteria following differential expression analysis. First, we removed any overlapping differentially expressed features. Following removal, we tested features for significant differences in residualized expression (Eq. (6)) for the opposite sex using Mann-Whitney U and removed significant features ($p < 0.05$).

### Subsampling male-only schizophrenia differential expression

For subsampling of the BrainSeq Consortium brain region analysis, we randomly sampled male individuals using the female sample sizes (121, 114, and 121 for the caudate nucleus, DLPFC, and hippocampus, respectively) and performed differential expression analysis (Eq. (6)) for schizophrenia. We performed this 1,000 times.

### Functional gene term enrichment analysis

We determined significant enrichment for gene sets using the gene set enrichment analysis (GSEA)[74,75]. Specifically, we performed GSEA with gseGO (gene ontology [GO] gene set database) from the clusterProfiler package[76] (version 4.6.2) and gseDGN (DisGeNET gene set database[77]) from the DOSE package[78] (version 3.24.2). We defined the gene set "universe" as all unique genes tested for differential expression. For gseGO, we set minimal gene set size (minGSSize) to 10, maximum gene set size (maxGSSize) to 500, and $p$ value cutoff to 0.05. For gseDGN, we set minGSSize to five and $p$ value cutoff to 0.05. We used the default settings for all other parameters.

For gene-term enrichment analysis for WGCNA modules, we used GOATOOLS Python package[79] (version 1.2.3) with the GO database and hypergeometric tests for enrichment and depletion as previously described[12]. Specifically, we converted GENCODE IDs to Entrez IDs using pybiomart (https://github.com/jrderuiter/pybiomart; version 0.2.0). With Entrez IDs, we applied enrichment analysis for each module. We performed multiple testing corrections using the Benjamini-Hochberg FDR method.

To measure GO term elements semantic similarity across brain regions, we used R package GOSemSim[80] (version 2.24.0) with the Wang method[38] and best-match average strategy.

### Sex interacting eQTL analysis in *cis* and region specificity

To identify sex-interacting cis-eQTL (si-eQTL) across the caudate nucleus, DLPFC, and hippocampus, we first separated out female and male individuals and, using PLINK 2.0, excluded variants with MAF less than 0.05 and variants with less than one allele by sex. To generate a common list, we overlapped these filtered variants, resulting in a total of 6,816,103 SNPs. Following SNP filtering, we performed eQTL analysis using tensorQTL[81,82] (version 1.0.7) for a sex-interaction model. We filtered low expression using the GTEx Python script, eqtl_prepare_expression.py, with modification for processing transcripts,

exons, and junctions. This retained features with expression estimates greater than 0.1 TPM in at least 20% of samples and aligned read count of six or more per brain region. Following low expression filtering, we performed TMM normalization on filtered counts using the GTEx Python script, rnaseqnorm.py (https://github.com/broadinstitute/gtex-pipeline/tree/master/qtl/src/rnaseqnorm.py). Following normalization, we implemented tensorQTL using an interaction linear regression model. To do this, we performed three major steps: (1) we adjusted expression for covariates (i.e., diagnosis, population stratification [SNP PCs 1–3], and expression PCs specific to brain region and feature); (2) selected *cis*-SNP using a mapping window of 0.5 Mb within the transcriptional start site of each feature; and (3) filtered SNPs based on an interaction MAF greater than or equal to 0.05 and the minor allele present in at least 10 samples.

To assess sharing across brain regions and to increase our power to detect si-eQTL effects, we used multivariate adaptive shrinkage in R (mashr[83]; version 0.2.57) as previously described[12]. mashr uses an empirical Bayes approach to learn patterns of similarity among conditions (e.g., brain regions) and then leverage these prior patterns to improve accuracy of effect size estimates. We obtained effect sizes and standard errors for these effect sizes from the tensorQTL interaction model results. To account for correlations among measurements across brain regions (i.e., overlapping sample donors), we used the estimate_null_correlation_simple function to specify a correlation structure prior to fitting the mash model. The mash model included both the canonical covariance matrices and data-driven covariance matrices learned from our data. We defined the data-driven covariance matrices as the top three PCs from the principal components analysis (PCA) performed on the significant signals (i.e., most significant nominal $p$ values by brain region). To learn the mixture weights and scaling for the si-eQTL effects, we initially fit the mash model with a random set (i.e., unbiased representation of the results) of the tensorQTL interaction model results (i.e., 5% for gene-SNP pairs and 1% for transcript-, exon-, and junction-SNP pairs). We next fitted these mixture weights and scaling to all of the si-eQTL results in chunks. We extracted posterior summaries and measures of significance (i.e., local false sign rate [lfsr]). We considered si-eQTL significant if the lfsr <0.05.

### Schizophrenia risk GWAS association

We downloaded the latest schizophrenia GWAS summary statistics with index and high-quality imputation SNPs as determined by Psychiatric Genomics Consortium (PGC version 3 [PGC3])[11]. Following download, we selected and converted PGC3 GWAS SNPs associated with BrainSeq Consortium SNPs as previously described[12]; specifically, we converted GWAS SNPs from hg19 to hg38 using PyLiftover, merged them with BrainSeq Consortium SNPs on hg38 coordinates, and matched alleles.

### Fine mapping and colocalization

We performed fine mapping and colocalization with gene level si-eQTL for the caudate nucleus, DLPFC, and hippocampus as previously described[12,84] with slight modification for priors. Briefly, we estimated priors from the tensorQTL nominal results with torus[85]. Following estimation of priors, we implemented DAP-G[86,87] (version 1.0.0) to generate posterior inclusion probabilities (PIP) that provide an estimate of the probability of a variant being causal for downstream colocalization with fastENLOC[88,89] (version 1.0). We applied fastENLOC with schizophrenia GWAS (PGC3)[11].

We visualized colocalization results using P-P plots and eQTL results from sex-only analysis. Specifically, we used tensorQTL as described above, to apply gene-level *cis*-eQTL analysis to female and male individuals separately without modification for sex interaction. We used a gene body window of 0.5 Mb, MAF greater than or equal to 0.01, and confounders generated from the *Sex-interacting eQTL*

*analysis in* cis *and region specificity*. We determined significance for the most highly associated variant per gene using empirical *p* values based on beta-distribution fitted with an adaptive permutation (1000–10,000). These *p* values were corrected for multiple testing across genes using Storey's *q* value. For each brain region, we generated P-P plots using sex-specific nominal and permutation results provided in Source Data for each significant colocalized gene identified (regional colocalization probability [RCP] > 0.5).

### General replication analysis
**Data download.** We downloaded differential expression results for sex differences from the supplemental materials for refs. 30–32. For sex differences in schizophrenia replication, we downloaded Qin et al. results[7]. For si-eQTL, we downloaded results from refs. 17–19,30.

For CommonMind Consortium replication of differential expression analysis, we downloaded differential expression results for sex differences from ref. 6, as well as normalized expression from Synapse (syn18103849). These results included two cohorts: NIMH HBCC (National Institute of Mental Health's Human Brain Collection Core) and MSSM-Penn-Pitt (MSSM: Mount Sinai NIH Brain Bank and Tissue Repository, Penn: University of Pennsylvania Brain Bank of Psychiatric Illnesses and Alzheimer's Disease Core Center, and Pitt: University of Pittsburgh NIH NeuroBioBank Brain and Tissue Repository) cohorts[90].

**Dosage compensation replication.** For dosage compensation replication, we calculated TPM using a mean insert size of 200. We computed the RXE as described above (*Dosage compensation*). We downloaded gene TPM from the GTEx v8 portal (https://www.gtexportal.org/home/datasets), as well as sample phenotype information. We computed RXE as described above (*Dosage compensation*).

**π1 replication analysis.** For π1 analysis, we initially selected all significant genes or eQTL (nominally significant, $p < 0.05$) from our results and compared them with results from an external dataset (i.e., CommonMind Consortium [CMC]). Using the *p* values from the external dataset, we calculated π0 with `qvalue` function from qvalue[91] (version 2.30.0). We calculated π1 with Eq. (10). The π1 statistic represents the fraction of effects shared between the two datasets.

$$\pi1 = 1 - \pi0 \tag{10}$$

### Graphics
We generated venn diagrams with *matplotlib_venn* (version 0.11.5) Python (version 3.8) package. We generated UpSet plots in R using ComplexHeatmap[92] (version 2.6.2). Unless otherwise stated, we generated box plots and scatterplots in R using ggpubr (version 0.4.0). We generated enrichment dot plots, enrichment heatmaps, and gene term enrichment plots using ggplot2[93]. To generate circos plots, we used circlize[94] (version 0.4.11) and ComplexHeatmap utilities in R. We used plotnine (version 0.12.1), a Python implementation of ggplot2, to generate enrichment heatmaps comparing public datasets with BrainSeq Consortium analysis, RXE scatterplots, and si-eQTL box plots. To generate rank-rank hypergeometric overlap (RRHO), we used the RRHO2[95,96] (version 1.0) and lattice packages in R.

### Reporting summary
Further information on research design is available in the Nature Portfolio Reporting Summary linked to this article.

## Data availability
The processed counts (GENCODE v25, hg38), linked de-identified phenotype information, and RNA quality metrics used in this study are publicly available from the BrainSeq Consortium Phase 2 and 3 data releases as RangedSummarizedExperiment R objects. The Phase 2 data (total RNA DLPFC and hippocampus) are available for download at http://eqtl.brainseq.org/phase2/. The Phase 3 data (total RNA caudate nucleus) are available for download at http://erwinpaquolalab.libd.org/caudate_eqtl/. The reprocessed data (GENCODE v41, hg38) is also available as an RangedSummarizedExperiment R object and is available as part of Source Data. Analysis-ready genotype data are available under restricted access to protect research subjects, access can be obtained through dbGaP accession phs000979.v3.p2. FASTQ files are also available under restricted access to protect research subject. For Phase 2 total RNA DLPFC and hippocampus, researchers can access to FASTQ files via the Globus collections jhpce#bsp2-dlpfc and jhpce#bsp2-hippo at https://research.libd.org/globus/. For Phase 3 caudate nucleus, researchers can obtain access to FASTQ files via dbGaP accession phs003495.v1.p1. PGC3 GWAS summary statistics are available at https://figshare.com/articles/dataset/scz2022/19426775. The nominal eQTL, predictive analysis, network analysis, and differential expression analysis generated in this study are provided in the Supplementary Information/Source Data file. They are also available at https://doi.org/10.5281/zenodo.7125279 or http://erwinpaquolalab.libd.org/3region_sex/. Source Data are provided at https://doi.org/10.5281/zenodo.7125279. Source data are provided with this paper.

## Code availability
All code and Jupyter Notebooks are available through GitHub at https://github.com/LieberInstitute/sex_differences_sz[97–104].

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

## Acknowledgements

The authors would like to extend their appreciation to the Offices of the Chief Medical Examiner of Washington DC, Northern Virginia, Kalamazoo Michigan, Santa Clara County, University of North Dakota, and Maryland for the provision of brain tissue used in this work. The authors also extend their posthumus appreciation to Dr. Llewellyn B. Bigelow and members of the LIBD Neuropathology Section for their work in assembling and curating the clinical and demographic information and organizing the Human Brain Tissue Repository of the Lieber Institute. Finally, the authors gratefully acknowledge the families that have donated this tissue to advance our understanding of psychiatric disorders. We also

thank Karen Ives for her editorial support. This work is supported by the Lieber Institute for Brain Development, the National Institutes of Health (NIH) T32 fellowship (T32MH015330) and K99 award (K99MD016964) to K.J.M.B., NIH R01 (R01MH123183) to L.C.-T., and a NARSAD Young Investigator Grant from the Brain & Behavior Research Foundation to J.A.E. R.A. would like to thank the Hopkins Office for Undergraduate Research (HOUR), Johns Hopkins University for their support through the Summer PURA program as well as the Albstein Research Scholarship for their support. This research was supported by the Intramural Research Program of the NIMH (NCT00001260, 900142).

## Author contributions

Conceptualization, K.J.M.B., R.A., A.C.M.P., and J.A.E.; Methodology, K.J.M.B., R.A., G.P., A.S.F., H.H.G., J.M.S., L.D.I., A.C.M.P., and J.A.E.; Software, K.J.M.B., R.A., G.P., A.S.F., H.H.G., W.S.U., and A.C.M.P.; Formal Analysis, K.J.M.B., R.A., G.P., A.S.F., and H.H.G.; Investigation, J.H.S. and T.M.H.; Data Curation, K.J.M.B., R.A., G.P., and J.E.K.; Writing—Original Draft, K.J.M.B., R.A., and J.A.E.; Writing—Review & Editing, K.J.M.B., R.A., G.P., A.S.F., H.H.G., L.C.-T., A.C.M.P., D.R.W., and J.A.E.; Visualization, K.J.M.B., R.A., L.D.I., and W.S.U.; Supervision, K.J.M.B., A.C.M.P., and J.A.E.; Project Administration, J.A.E.; Funding Acquisition, K.J.M.B., R.A., L.C.-T., D.R.W., and J.A.E.

## Competing interests

D.R.W. serves on the Scientific Advisory Boards of Sage Therapeutics and Pasithea Therapeutics. J.E.K. is a member of a drug monitoring committee for an antipsychotic drug trial for Merck. All other authors declare no competing interests.
