## [Peer Review File · Nature Communications]

Sex affects transcriptional associations with schizophrenia across the dorsolateral prefrontal cortex, hippocampus, and caudate nucleusREVIEWER COMMENTS

Reviewer #1 (Remarks to the Author):

Benjamin et al. analyze the impact of sex on brain-region specific expression across hundreds of post-mortem schizophrenia and control samples collected by the BrainSeq consortium. The authors identify hundreds of sex-biased genes on autosomes and sex chromosomes alike, with most detected in the caudate nucleus.

The main strength of this study lies in the large number of samples analyzed, going well beyond similar studies with more limited brain-region specific sample sizes. The methodological approach is sound and well-suited for this exploratory analysis (linear regression, differential expression and WGCNA). Additional strengths include a permutation analysis to determine power for detecting sex differences in differentially expressed gene (DEG) counts in specific brain regions. Indeed an excess of male samples does increase power for detecting such region-specific differential expression and sex-interacting eQTLs.

The manuscript is well written and the figures clearly presented. There are some minor methodological quibbles that could be easily addressed:

- 1.) outdated Gencode v25 (released early 2016)
- 2.) Gene set enrichment (GSEA) may be more appropriate than hypergeometric enrichment testing, because no hard significance cutoffs are used to define DEGs
- 3.) Could the replication analysis could be done via module preservation of WGCNA modules?

Some additional biological questions came up as well:

- 1.) Were the authors expecting a specific brain region to be most reflective of schizophrenia? Do these results square with what is currently known about this question?
- 2.) Why the focus on ANK3 over the other 13 identified genes? Do the results point to a brain-region specific "injury" incurred by ANK3 dysregulation?
- 3.) The relative resistance of females to schizophrenia is interesting. I was wondering whether the authors could leverage additional information on age at diagnosis to infer whether such protection is mediated by brain-region specific expression or estrogen levels? A quick google search indicated estrogen is sometimes used for treating schizophrenic females and low estrogen (post-menopause) increases susceptibility to treatment-resistant schizophrenia?

The authors regressed out age as a factor, but could they preserve age to address this question (ie. whether estrogen-dependent gene expression decreases with age and overlaps with sex-differential schizophrenia DEGs).

Overall, this is a well-presented and thorough study of how sex and schizophrenia interact in 3 different brain regions.

Reviewer #2 (Remarks to the Author):

Benjamin et. al. analyzed 1170 samples across 504 individuals in caudate nucleus (n=399), dorsolateral prefrontal cortex (DLPFC; n=377), and hippocampus (n=394). The study systematically investigates 1) sex specific changes in three brain regions, 2) sex-brain region-specific and associated X-chromosome inactivation, 3) schizophrenia-sex specific differences and 4) sex-dependent eQTL analysis. Overall, these results provide novel insights into sex differences and have identified sex-

specific schizophrenia genes and provides the annotation of si-eQTL in the human DLPFC, hippocampus and caudate nucleus. Additionally, study showed that X-chromosome dosage was significantly decreased in the hippocampus of male and female schizophrenic individuals. It's a nice comprehensive analyses of sex-specific differences in each brain region.

Minor Comments:

- 1) Can you add how many up and down regulated genes were found for each brain region in section "Sex-specific expression across the caudate nucleus, DLPFC, and hippocampus"?
- 2) WGCNA analysis for all different types of analysis showed $R < .01$. Why is it important to have analysis? It seems to provide no insights.
- 3) Could you please add MAGMA enrichment analysis of DEGs for all sections for brain and non-brain related traits? This can be done separately for up and downregulated genes for each brain region.
- 4) Could you please add pi1 stats for sex-specific schizophrenia DEGs and schizophrenia specific DEGs from CMC paper (Fromer et.al. 2018)?
- 5) Also, could you please add pi1stats for si-eQTL and eQTL analysis for each brain region?
- 6) Does the 14 si-QTL co-localized genes have sex specificity in schizophrenia and controls patients?
- 7) Equation 2 does not have interaction term of sex and region.

Reviewer #3 (Remarks to the Author):

Benjamin & Arora et al. in this work use sequencing and genotype data from BrainSeq Consortium to explore sex-differences in the caudate nucleus, hippocampus, and dorsolateral PFC in the context of schizophrenia. Authors identify region- and brain-wide sex differences as well as schizophrenia-associated expression and eQTLs changes. Overall, I appreciated the rigorous take on analysis and the attempt at creating multifaceted compiled results with a focus on sex differences in the context of disease. Their results complement already available studies on sex differences in schizophrenia and other psychiatric disorders and contribute to filling the gap of sex-dependent changes in psychiatric disease. Additionally, their work further supports and proposes the importance of taking sex into consideration in the field, giving it a broad interest for the field.

I do, however, have some suggestions I would like the authors to address:

Minor:

- I haven't been able to find a detailed description of how the differential gene expression analysis differed between gene, transcript, exon and junction-level analyses.
- In Fig. 1E, the authors explore the relationships between regions. Why relying on Pearson correlation rather than RRHO that authors use later in the manuscript? Additionally, the correlation dramatically increases when using only DEGs. Is the correlation still present when using all genes except the DEGs? Is the correlation indeed present regardless of the thresholding in significance (non-DEG vs DEG) for the genes, or is it only driven by the DEGs?
- Authors in lines 426-428 highlight three genes that they deem relevant, but it is not clear why these genes specifically are highlighted or worth of highlight. Additionally, it should be pointed out if these genes somehow stand out from a statistical point of view.
- Equation 9 (lines 237-238) is missing.
- In the manuscript it is possible to find some incomplete sentences, or sentences with meanings difficult to understand, e.g., line 384 "When we examined the functional relevance of these modules, we found significant enrichment (hypergeometric test, $FDR < 0.05$) for terms related to the modules, [...]", line 539-540 "The five most significant female-specific schizophrenia up- and down-regulated genes for caudate nucleus.", lines 599-600 "Here, we found the majority (814 [84%]) of eGenes were shared across brain regions (Fig. 4A), which was also on the transcript, exon, and junction level."
- Authors identify the majority of eGenes on autosomes rather than on the X chromosome. How much

of these results are influenced by the distribution of the eQTLs analyzed between autosomes and the X chromosome? Since the former is much more extensive, I would expect to have a higher amount of variants analyzed.

- In Fig. 3 the differential transcriptome of male and female caudate shows a highly significant correlation, and a moderate correlation for DLPFC transcriptomes. So, how could the authors explain that the correlation between DLPFC and the caudate was very strong in females but completely absent in males?
- I am not sure I understand why in figure S5 the highest autosomal DEGs are not shown. Authors could make the figure and/or the legend clearer.
- In Fig. 4B the colour scale is deceiving since the lightest colour indicates the highest overlap. Inverting the scale like in Fig. S26 (dark colour corresponding to high percentage and vice versa) would improve readability.
- In Fig. S9 groups of overlap are coloured differently, but no legend is provided for this colour fill.
- In Fig. S14 the colour scheme is quite confusing since the same colour pairing is used for different groups.
- Line 411 "male-bias" is in a different font size.
- Authors are not always consistent in their notations: $-\log_{10}$ vs $-\text{Log}_{10}$, p-value vs p adjust.

Major

- Authors sometimes seem to not be able to highlight the results of their study and the novelty they collected. For example, the WGCNA results are only briefly described, with no mention of possible relevant genes and no further elaboration in the discussion section. Authors seem to heavily rely on supplementary data, but a more critical and comprehensive description of this data already in the results and discussion section would benefit the reader.
- Authors mention that previous studies on sex-differences in schizophrenia have solely focused on prefrontal cortex (lines 45-46). It would be important, at least in the discussion, to address how different or similar the results obtained in this study, on new regions, relate to the previously available datasets.
- It is my understanding that authors analyze unique-associated genes to the eQTLs they explored, however, they do not attempt to relate their eQTL results with their DEGs results. Are any of these eQTL related to the genes previously identified for being differentially expressed by condition?
- In the results and discussion section authors highlight ANK3 as a gene of particular interest and relevance among all their results. Has the sex-difference in ANK3 expression level been validated somehow? Do other studies or other datasets could corroborate their results? Looking into previously published sequencing data in either human or mouse models could be used as validation to strengthen their final message.

Response to reviewers' questions

We are grateful to the reviewers for their thoughtful critiques and suggestions, which have substantially improved the manuscript. We are also pleased that the reviewers appreciated the scope and methodological approach to examining transcriptional sex differences associated with schizophrenia. We thank the editor's for understanding of the substantial effort undertaken to reprocess 1170 samples. Here, we address all comments and suggestions from the reviewers.

Remarks to the authors:

Reviewer #1 (Remarks to the Authors):

Benjamin et al. analyze the impact of sex on brain-region specific expression across hundreds of post-mortem schizophrenia and control samples collected by the BrainSeq consortium. The authors identify hundreds of sex-biased genes on autosomes and sex chromosomes alike, with most detected in the caudate nucleus.

The main strength of this study lies in the large number of samples analyzed, going well beyond similar studies with more limited brain-region specific sample sizes. The methodological approach is sound and well-suited for this exploratory analysis (linear regression, differential expression and WGCNA). Additional strengths include a permutation analysis to determine power for detecting sex differences in differentially expressed gene (DEG) counts in specific brain regions. Indeed an excess of male samples does increase power for detecting such region-specific differential expression and sex-interacting eQTLs.

The manuscript is well written and the figures clearly presented.

We appreciate the very positive comments and summarization of our study from **Reviewer #1**. Additionally, **Reviewer #1** had several minor quibbles and suggestions for improvement.

Reviewer #1 comment: *outdated Gencode v25 (released early 2016)*

Response: We agree that GENCODE v25 is outdated. As a reprocessing analysis, we originally used the preprocessed official releases of the BrainSEQ consortium data (GENCODE v25). As we have access to the original FASTQ files, we have re-aligned the data to GENCODE v41 (released July 2022). With this major change, we have re-analyzed the data with the new, harmonized, counts. As a result, we had several changes. The difference in detected features is expected as we use different covariates based on specific alignment results from GENCODE v41. We determine covariates by evaluating potential confounders for expression or sex.

Even so, when we compared ensembl IDs – gene IDs that do not vary with GENCODE version – between effect sizes of our main DE models (sex, region interaction with sex, and sex-specific schizophrenia models) between GENCODE v25 and v41, we found significant correlations (Spearman's correlation; all $\rho > 0.49$, all $p\text{-value} < 0.017$; **Fig. R1**, **Fig. R2**, and **Fig. R3**), which increased to over 86% when removed the lowest correlation (pairwise comparison between DLPFC and hippocampus). Furthermore, when we compared significant genes across these DE models, our correlation of effect sizes increased to over 97% (Spearman's correlation; r^2).

We summarize these changes below.

1. A decrease in the number of detected features:
 - a. Unique sex DEGs (878 to 831)
 - b. Unique male-specific schizophrenia DEGs
 - i. 1858 to 1130 for the caudate nucleus
 - ii. 122 to 5 for the DLPFC. We attribute the lower detection for male-specific schizophrenia to a decrease in power with the increase in genes tested as the effect sizes between annotations are nearly identical (Spearman's correlation; $\rho = 0.99$, p-value ~ 0 ; **Fig. R3B**).
 - c. Unique eGenes (974 to 704)
 - d. Colocalized genes (14 to nine)
2. A decrease in replication rates primarily due to our replication datasets (GTEx) aligned to GENCODE v26. The lowest sex-specific replication decreased from 66% to 62% in all brain regions except for the GTEx cerebellum and anterior cingulate cortex (**Fig. S9**).
3. An increase in the number of detected features for:
 - a. Sex-interacting schizophrenia junctions from 15 to 148 across brain regions. We attribute the increase of detected junctions to better annotation of exon-exon junctions in the newer version of GENCODE v41.
 - b. Unique female-specific schizophrenia DEGs in the caudate (124 to 194)
 - c. Unique male-specific schizophrenia DEGs in the hippocampus (104 to 149)
4. An increase in autosomal predictive power for sex. With the new version of GENCODE v41, autosomal predictive power drastically increased from less than 60% using the 100 more significant DEGs to greater than 88% using the ten most significant DEGs. This was primarily driven by one pseudogene (*RPS10P3* [ribosomal protein S10 pseudogene 3]) shared across brain regions, which was not previously found to be significantly different using GENCODE v25.

Reviewer #1 comment: *Gene set enrichment (GSEA) may be more appropriate than hypergeometric enrichment testing, because no hard significance cutoffs are used to define DEGs*

Response: We thank the review with this suggestion and have replaced hypergeometric enrichment testing with GSEA. This has increased the clarity of the enrichment analysis.

Reviewer #1 comment: *Could the replication analysis could be done via module preservation of WGCNA modules?*

Response: We thank the review for this question. Module preservation is generally used to assess potential conserved networks. As WGCNA module preservation analysis compares shared networks between two groups, this limits the ability as a replication analysis. Specifically, if a network is unique to one group this will be missed. Even so, module preservation can be used as an orthogonal approach and to compare shared networks. Therefore, we have added module preservation analysis for our sex-specific expression analysis and sex-specific schizophrenia analysis. As a result, we found that 1) shared network modules between female and male individuals were well preserved ($Z > 10$) similar to others⁴⁴; and 2) one module was significantly not preserved ($Z < 10$) for the DLPFC for individuals with schizophrenia (between female and male individuals). We have updated the manuscript to include these results and analyses.

Reviewer #1 comment (biological questions): *Were the authors expecting a specific brain region to be most reflective of schizophrenia? Do these results square with what is currently known about this question?*

Response: From our previous work of schizophrenia in the postmortem brain, we were expecting to see the caudate nucleus to show significantly more dysregulated features than the other two brain regions – due primarily to antipsychotic drugs targeting this region². Additionally, we expected the DLPFC and hippocampus to show similar patterns of expression based on other work on these regions³. We were not sure, however, if there would be a specific brain region with more female- or male-specific DEGs. Our results do support this other work and we have updated the discussion to include this information.

Reviewer #1 comment (biological questions): *The relative resistance of females to schizophrenia is interesting. I was wondering whether the authors could leverage additional information on age at diagnosis to infer whether such protection is mediated by brain-region specific expression or estrogen levels? A quick google search indicated estrogen is sometimes used for treating schizophrenic females and low estrogen (post-menopause) increases susceptibility to treatment-resistant schizophrenia?*

The authors regressed out age as a factor, but could they preserve age to address this question (ie. whether estrogen-dependent gene expression decreases with age and overlaps with sex-differential schizophrenia DEGs).

Response: We thank the reviewer for this very insightful question. Unfortunately, we do not have first episode data in our patient information. Additionally, review of our differential analysis also did not show a significant differences for any of the sex hormone genes (*AR* [androgen receptor], *ESR1* [estrogen receptor 1], *ESR2* [estrogen receptor 2], and *PGR* [progesterone receptor]) across the brain. When we examined a targeted analysis of estrogen-dependent gene expression association with age and diagnosis status, we found only nominally significant association with *ESR2* (p-value = 0.016 [FDR = 0.098]; **Fig. S23**) for female individuals and *PGR* (p-value = 0.013 [FDR = 0.16]; **Fig. S24**) for male individuals in the hippocampus. This may be due to the small sample size and uneven classes (i.e., female and male). While we cannot say for sure if there is a significant association with sex hormones and schizophrenia in the brain, we believe this possibility warrants the inclusion of the sex hormone analysis to this manuscript. As such, we have included this in the results and discussion.

Reviewer #1 comment: *Overall, this is a well-presented and thorough study of how sex and schizophrenia interact in 3 different brain regions.*

Response: We thank the reviewer for this comment.

Reviewer #1 asked “*Why the focus on ANK3 over the other 13 identified genes? Do the results point to a brain-region specific ‘injury’ incurred by ANK3 dysregulation?*” and **Reviewer #3** commented “*In the results and discussion section authors highlight ANK3 as a gene of particular interest and relevance among all their results. Has the sex-difference in ANK3 expression level been validated somehow? Do other studies or other datasets could corroborate their results? Looking into previously published sequencing data in either human or mouse models could be used as validation to strengthen their final message.*”

We initially focused on *ANK3* due to its link to schizophrenia and role as a marker for early life stress. We believe this might be an important gene to discuss in more detail due to the vulnerability of male individuals to stress in utero¹. While *ANK3* still may play an important role in sex-specific schizophrenia, we were unable to replicate its sex-specific expression in the caudate nucleus in GTEx (local false sign rate [lfsr] = 0.079) or find additional indications of sex-specific schizophrenia dysregulation within our schizophrenia analyses (i.e., sex-interacting schizophrenia and sex-specific

schizophrenia). We also did not find a sex-interacting eQTL for *ANK3*. Therefore, we have deemphasized this finding from the main text.

Fig. R1. Significant correlation between annotations for sex DE model. Scatterplot comparing effect size (log fold-change) of all genes tested shared between Gencode version 25 and 41. A fitted trend line is presented in blue as the mean values +/- standard deviation. The standard deviation is shaded in light blue.

Fig. R2. Significant correlation between annotations for the brain region interacting with sex DE model. Scatterplot comparing effect size (log fold-change) of all genes tested in GENCODE version 41 and significant DEGs (adjusted p-value < 0.05) in GENCODE version 25. A fitted trend line is presented in blue as the mean values +/- standard deviation. The standard deviation is shaded in light blue.

Fig. R3. Significant correlation between annotations for sex-specific schizophrenia DE model. Scatterplot comparing effect size (log fold-change) of all genes tested in GENCODE version 41 and significant DEGs (adjusted p -value < 0.05) in GENCODE version 25 for **A.** female-specific comparison and **B.** male-specific schizophrenia analyses. A fitted trend line is presented in blue as the mean values \pm standard deviation. The standard deviation is shaded in light blue.

Fig. S9. High overlap of sex-specific differentially expressed genes (DEGs) across multiple datasets and brain regions. Heatmaps comparing BrainSeq Consortium brain regions that show the ratio of DEGs overlap with **A.** Trabzuni *et al.* brain regions, **B.** Mayne *et al.* meta analysis with significant expression in at least one brain region using all chromosomes, only autosomes, or no Y chromosomes, and **C.** Gershoni and Pietrokovski brain regions. Abbreviations: CRBL: cerebellum, FCTX: frontal cortex, HIPP: hippocampus, HYPO: hypothalamus, MEDU: medulla, OCTX: occipital cortex, PUTM: putamen, SNIG: substantia nigra, TCTX: temporal cortex, THAL: thalamus, and WHMT: white matter.

Fig. S23. Nominal significant interaction of age and diagnosis in the hippocampus for *ESR2* for female individuals. Scatterplot of residualized expression showing correlation with age as a function of diagnosis status (control [CTL] in red and schizophrenia [SZ] in blue). A fitted trend line is presented as the mean values +/- standard deviation separated by diagnosis status (control in red and schizophrenia in blue). The standard deviation is shaded in by diagnosis status (control in red and schizophrenia in blue).

Fig. S24. No significant interaction of age and diagnosis in the hippocampus for *ESR2* for male individuals. Scatterplot of residualized expression showing no significant correlation with age as a function of diagnosis status (control [CTL] in red and schizophrenia [SZ] in blue). A fitted trend line is presented as the mean values +/- standard deviation separated by diagnosis status (control in red and schizophrenia in blue). The standard deviation is shaded in by diagnosis status (control in red and schizophrenia in blue).

Reviewer #2 (Remarks to the Authors):

Benjamin et. al. analyzed 1170 samples across 504 individuals in caudate nucleus (n=399), dorsolateral prefrontal cortex (DLPFC; n=377), and hippocampus (n=394). The study systematically investigates 1) sex specific changes in three brain regions, 2) sex-brain region-specific and associated X-chromosome inactivation, 3) schizophrenia-sex specific differences and 4) sex-dependent eQTL analysis. Overall, these results provide novel insights into sex differences and have identified sex-specific schizophrenia genes and provides the annotation of si-eQTL in the human DLPFC, hippocampus and caudate nucleus. Additionally, study showed that X-chromosome dosage was significantly decreased in the hippocampus of male and female schizophrenic individuals. It's a nice comprehensive analyses of sex-specific differences in each brain region.

We thank the reviewer for these very positive comments and summarization of our work. We address **Reviewer #2's** minor comments below.

Reviewer #2 comment: *Can you add how many up and down regulated genes were found for each brain region in section "Sex-specific expression across the caudate nucleus, DLPFC, and hippocampus"?*

Response: We agree that quantifying the number of up- and down-regulated genes would improve clarity and have updated the main text to include up- and down-regulated genes for each brain region in the sex-specific expression section (excerpt below).

We observed 831 unique, DEGs (FDR < 0.05; **Fig. 1A**) between the sexes across the caudate nucleus (n=689 DEGs [279 upregulated in females; 410 upregulated in males]), DLPFC (n=256 [99 upregulated in females; 157 upregulated in males]), and hippocampus (n=147 [64 upregulated in females; 83 upregulated in males]).

Reviewer #2 comment: *WGCNA analysis for all different types of analysis showed $R < .01$. Why is it important to have analysis? It seems to provide no insights.*

Response: We added the network analysis as an orthogonal approach for our differential analysis. However, we agree that the WGCNA adds marginal insight to the manuscript as currently written. We have updated these sections to highlight pathway enrichment replication for differential expression analysis. Additionally, we have added preservation network analysis to improve additional insights of sex-specific co-expression networks. We believe this improves the reader's insight from these analyses.

Reviewer #2 comment: *Could you please add MAGMA enrichment analysis of DEGs for all sections for brain and non-brain related traits? This can be done separately for up and downregulated genes for each brain region.*

Response: We thank the reviewer for this suggestion. We have added MAGMA enrichment analysis of DEGs for all sections using brain (attention deficit hyperactivity disorder, anorexia, autism spectrum disorder, bipolar disorder, body mass index, major depressive disorder, and schizophrenia) and non-brain (height, monocyte, neutrophil, basophil, and eosinophil) related traits. For our DEG models that did not include an interaction term (sex and sex-specific schizophrenia), we performed this MAGMA enrichment analysis separately for upregulated and downregulated DEGs for each brain region as suggested. For our brain region and sex interacting model, we applied MAGMA for each pairwise interacting (caudate vs DLPFC, caudate vs hippocampus, and DLPFC vs hippocampus).

When we performed this analysis, we found no significantly enriched traits for either female- or male-biased DEGs and brain region and sex interacting analysis (**Data S3**). For sex-specific schizophrenia analysis, we found significant enrichment of schizophrenia and immune-related blood cell types traits. Specifically, we found enrichment of schizophrenia and neutrophils traits for male-specific DEGs upregulated in individuals with schizophrenia for the caudate nucleus and hippocampus (**Data S3**). We also found significant enrichment of the schizophrenia trait for female-specific DEGs upregulated in individuals with schizophrenia for the caudate nucleus (**Data S3**). In contrast, we found significant enrichment of basophil and eosinophils for male-specific DEGs downregulated in individuals with schizophrenia for the DLPFC (**Data S3**).

Reviewer #2 comment: *Could you please add pi1 stats for sex-specific schizophrenia DEGs and schizophrenia specific DEGs from CMC paper (Fromer et.al. 2018)?*

Response: We agree that pi1 statistics would improve the rigor to our comparison analysis. While neither we nor the CMC sex-specific manuscript⁴ found significant genes after correcting for multiple testing for sex-interacting schizophrenia DEGs, we have added pi1 statistic results for the two CMC cohorts using nominally significant p-values (**Table S5**).

Reviewer #2 comment: *Also, could you please add pi1 stats for si-eQTL and eQTL analysis for each brain region?*

Response: We have added pi1 statistics for si-eQTL/eQTL analysis for a pairwise comparison of brain regions (**Fig. S28**).

Reviewer #2 comment: *Does the 14 si-QTL co-localized genes have sex specificity in schizophrenia and controls patients?*

Response: We now include information on the si-eQTL colocalized genes for sex specificity in schizophrenia and control patients.

After reprocessing the data on GENCODE v41, we now have nine si-eQTL colocalized genes. We identified a noncoding RNA (*ENSG00000287222*) as a sex-specific schizophrenia DEG (male-specific, adjusted p-value = 0.026) in the caudate nucleus. This gene was shared with the hippocampus, however, we did not identify it as also a sex-specific schizophrenia DEG for the hippocampus. While we did not observe any tissue-specific overlap between the other eight colocalized genes and sex-specific schizophrenia DEGs – potentially due to low number of identified DEGs, we did observe one additional colocalized gene (*FURIN*; male-specific, adjusted p-value = 0.023) as a sex-specific schizophrenia DEG in the caudate nucleus. Interestingly, we found *ACE* to be downregulated in schizophrenia for both sexes in the caudate nucleus. We believe that while si-eQTL appears to be recurrent across the brain, their sex-specific expression may be tissue specific.

Reviewer #2 comment: *Equation 2 does not have interaction term of sex and region.*

Response: We thank the reviewer for catching this error and have fixed the equation to include the interaction term.

Fig. S28. High level of replication of si-eQTL across brain regions. Histogram of significant si-eQTL ($lfsr < 0.05$) from nominal p-values generated from **A.** female-only and **B.** male-only eQTL analyses. π_1 (π_1) statistic annotated on histograms.

Table S5. Summary of π_1 statistic for sex-specific schizophrenia nominally significant DEGs (p-value < 0.05) between BrainSeq (caudate nucleus, DLPFC, and hippocampus) and CMC DLPFC by cohort.

CMC DLPFC cohort	BrainSeq Region	π_1 statistic
MSSM-Penn-Pitt	Caudate nucleus	0
	DLPFC	0
	Hippocampus	0
NIMH HBCC	Caudate nucleus	0.07
	DLPFC	0.51
	Hippocampus	0

$$E(Y) = \beta_0 + \beta_1 Sex * \beta_2 Region + \beta_3 Diagnosis + \beta_4 Age + \beta_5 MitoRate + \beta_6 rRNArate + \beta_7 TotalAssignedGenes + \beta_8 RIN + \beta_9 OverallMappingRate + \beta_{10} Mean3Bias + \sum_{i=1}^3 \eta_i snpPC_i$$

Equation 2

Data S3. BrainSeq_MAGMA_enrichment_analysis.xlsx: Excel file of magma enrichment results of all DEGs (i.e., sex-specific, sex interacting with brain region, and sex-specific schizophrenia) separated by direction of effect across the caudate nucleus, DLPFC, and hippocampus.

Reviewer #3 (Remarks to the Authors):

Benjamin & Arora et al. in this work use sequencing and genotype data from BrainSeq Consortium to explore sex-differences in the caudate nucleus, hippocampus, and dorsolateral PFC in the context of schizophrenia. Authors identify region- and brain-wide sex differences as well as schizophrenia-associated expression and eQTLs changes. Overall, I appreciated the rigorous take on analysis and the attempt at creating multifaceted compiled results with a focus on sex differences in the context of disease. Their results complement already available studies on sex differences in schizophrenia and other psychiatric disorders and contribute to filling the gap of sex-dependent changes in psychiatric disease. Additionally, their work further supports and proposes the importance of taking sex into consideration in the field, giving it a broad interest for the field.

We thank the reviewer for this excellent summary and positive feedback on our work. Below we address the reviewer's major and minor comments.

Minor:

Reviewer #3 comment: *I haven't been able to find a detailed description of how the differential gene expression analysis differed between gene, transcript, exon and junction-level analyses.*

Response: We agree that a more detailed description of isoform-level results, in perspective of their gene level counterpart, would improve the manuscript. We have now added a brief discussion of isoform-level DE analysis for the sex-specific and brain region-interacting with sex DE analyses. We omit this additional analysis for the sex-specific schizophrenia DE analysis as they were underpowered for female-specific schizophrenia DE analysis.

For sex-specific isoform-level DE analysis, we identified an additional 859 unique genes associated with a differentially expressed transcript, exon, or exon-exon junction (**Fig. S4**). For brain region interaction with sex DE analysis, we identified more than double unique DEGs (1303 [775 isoform only], 198 [127 isoform only], and 23 [18 isoform only], for caudate nucleus vs DLPFC, caudate nucleus vs hippocampus, and DLPFC vs hippocampus, respectively; **Fig. S14**).

Reviewer #3 comment: *In Fig. 1E, the authors explore the relationships between regions. Why relying on Pearson correlation rather than RRHO that authors use later in the manuscript? Additionally, the correlation dramatically increases when using only DEGs. Is the correlation still present when using all genes except the DEGs? Is the correlation indeed present regardless of the thresholding in significance (non-DEG vs DEG) for the genes, or is it only driven by the DEGs?*

Response: We apologize for the lack of clarity. In the original **Fig. 1E**, we present the non-parametric Spearman correlation of the t-statistic all genes regardless of significance. To help with additional clarity, we have switched to effect size in the main figure and added RRHO to the supplementary. We find the correlation is still very strong with all genes, which can be seen in the new **Fig. 1E** and **RRHO Fig. S12**. We believe with the inclusion of RRHO that the correlation is indeed present regardless of the thresholding of significance.

Reviewer #3 comment: *Authors in lines 426-428 highlight three genes that they deem relevant, but it is not clear why these genes specifically are highlighted or worth of highlight. Additionally, it should be pointed out if these genes somehow stand out from a statistical point of view.*

Response: We agree with the reviewer that the significance of these genes were not clearly explained. Originally, these genes were selected based on being 1) protein coding and 2) brain-region specific.

However, this was indeed an arbitrary selection. As such, we have replaced the discussion of these genes with pathway enrichment analysis, which we believe to be more interpretable and relevant to the manuscript. As a result, we have replaced the original three genes with example sex-biased, brain region-specific DEGs to **Fig. 1F**. These genes are just examples of genes that show a significant interaction between sex and brain region.

Reviewer #3 comment: *Equation 9 (lines 237-238) is missing.*

Response: We apologize for the missing equation and have made sure to include it in the manuscript.

Reviewer #3 comments:

In the manuscript it is possible to find some incomplete sentences, or sentences with meanings difficult to understand, e.g., line 384 “When we examined the functional relevance of these modules, we found significant enrichment (hypergeometric test, FDR < 0.05) for terms related to the modules, [...]”, line 539-540 “The five most significant female-specific schizophrenia up- and down-regulated genes for caudate nucleus.”, lines 599-600 “Here, we found the majority (814 [84%]) of eGenes were shared across brain regions (Fig. 4A), which was also on the transcript, exon, and junction level.”

Authors are not always consistent in their notations: $-\log_{10}$ vs $-\text{Log}_{10}$, p -value vs p adjust.

Response: We thank the reviewer for finding these errors. We have contracted a copy editor to review the manuscript for these mistakes, including consistency.

Reviewer #3 comment: *Authors identify the majority of eGenes on autosomes rather than on the X chromosome. How much of these results are influenced by the distribution of the eQTLs analysed between autosomes and the X chromosome? Since the former is much more extensive, I would expect to have a higher amount of variants analyzed.*

Response: We agree that our observation might be influenced by distribution. Indeed, when we compared eGenes based on ratio, we found that there was no significant enrichment or depletion of si-eQTL located on the X chromosome (Kolmogorov-Smirnov test, p -value=0.89). We have updated the manuscript to address this.

Reviewer #3 comment: *In Fig. 3 the differential transcriptome of male and female caudate shows a highly significant correlation, and a moderate correlation for DLPFC transcriptomes. So, how could the authors explain that the correlation between DLPFC and the caudate was very strong in females but completely absent in males?*

Response: For the DLPFC, the limited observable overlap is apparent in the opposing enrichment patterns between DLPFC in females (**Fig. 3A**) and males (**Fig. 3B**); while female transcriptional signatures associated with genes upregulated in schizophrenia, male transcriptional signatures associated with genes downregulated. This is also the case for the hippocampus. We have updated the results to include this explanation.

Reviewer #3 comment: *I am not sure I understand why in figure S5 the highest autosomal DEGs are not shown. Authors could make the figure and/or the legend clearer.*

Response: We apologize for the lack of clarity. We did not show hippocampus N autosome DEGs because the highest N included all of the DEGs. After reprocessing the data, we have updated the plot

to include the top 10 most significant DEGs instead (**Fig. S6**). We have also updated the figure and figure legend to improve clarity.

Reviewer #3 comment: *In Fig. 4B the colour scale is deceiving since the lightest colour indicates the highest overlap. Inverting the scale like in Fig. S26 (dark colour corresponding to high percentage and vice versa) would improve readability.*

Response: We agree that a color change would improve readability and have made this change to **Fig. 4B**.

Reviewer #3 comment: *In Fig. S9 groups of overlap are coloured differently, but no legend is provided for this colour fill.*

Response: We thank the reviewer for catching this accidental omission. We have updated the figure legends in the main and supplementary information (**Figs. 1, 4** and **Figs. S10, S27**) for all UpSet plots to provide this detail on the color of the overlaps.

Reviewer #3 comment: *In Fig. S14 the colour scheme is quite confusing since the same colour pairing is used for different groups.*

Response: We apologize for any confusion. We use different color schemes for **Fig. S14A** (now **Fig. S18A**) and **Fig. S14B** (now **Fig. S18B**) so that it is clear the plot in **Fig. S16A** is a comparison between female and male, while **Fig. S18B** is a comparison between control and schizophrenia. However, this figure is not very clear as the x-axis is very small. We have increased the size of the x-axis so that it is more apparent.

Reviewer #3 comment: *Line 411 “male-bias” is in a different font size.*

Response: We thank the reviewer for catching this error. We have fixed all font sizes in the main text so that they are consistent.

Major:

Reviewer #3 comment: *Authors sometimes seem to not be able to highlight the results of their study and the novelty they collected. For example, the WGCNA results are only briefly described, with no mention of possible relevant genes and no further elaboration in the discussion section. Authors seem to heavily rely on supplementary data, but a more critical and comprehensive description of this data already in the results and discussion section would benefit the reader.*

Response: We agree with the reviewer that a more elaborate interpretation and discussion of our results would benefit the readers. We have increased our discussion to include more discussion and interpretation to address this issue. Additionally, we have added more interpretation of results within each results section to remove the burden of understanding our work from the reader.

Reviewer #3 comment: *Authors mention that previous studies on sex-differences in schizophrenia have solely focused on prefrontal cortex (lines 45-46). It would be important, at least in the discussion, to address how different or similar the results obtained in this study, on new regions, relate to the previously available datasets.*

Response: We agree with the reviewer that it is important to discuss why the prefrontal cortex has been the focus of schizophrenia research. As such, we have included this in the discussion (paragraph 4). Additionally, we have added additional clarity on why we examined other brain regions.

Reviewer #3 comment: *It is my understanding that authors analyze unique-associated genes to the eQTLs they explored, however, they do not attempt to relate their eQTL results with their DEGs results. Are any of these eQTL related to the genes previously identified for being differentially expressed by condition?*

Response: We thank the reviewer for their insightful suggestion and agree that additional exploration of the si-eQTL with our DEG results would improve the manuscript. As such, we have updated the manuscript to include this information. We did not find any significant enrichment (Fisher's exact test, p -value > 0.10) of si-eQTL associated eGenes with sex-specific DEGs. We did find some overlap with sex-specific schizophrenia DEGs and the colocalized si-eQTL.

Reviewer #3 commented “*In the results and discussion section authors highlight ANK3 as a gene of particular interest and relevance among all their results. Has the sex-difference in ANK3 expression level been validated somehow? Do other studies or other datasets could corroborate their results? Looking into previously published sequencing data in either human or mouse models could be used as validation to strengthen their final message.*” **Reviewer #1** had a similar question and asked “*Why the focus on ANK3 over the other 13 identified genes? Do the results point to a brain-region specific ‘injury’ incurred by ANK3 dysregulation?*” and

We initially focused on *ANK3* due to its link to schizophrenia and role as a marker for early life stress. We believe this might be an important gene to discuss in more detail due to the vulnerability of male individuals to stress in utero¹. While *ANK3* still may play an important role in sex-specific schizophrenia, we were unable to replicate its sex-specific expression in the caudate nucleus in GTEx (local false sign rate [lfsr] = 0.079) or find additional indications of sex-specific schizophrenia dysregulation within our schizophrenia analyses (i.e., sex-interacting schizophrenia and sex-specific schizophrenia). We also did not find a sex-interacting eQTL for *ANK3*. Therefore, we have deemphasized this finding from the main text.

Fig. 1. Sex-biased expression across the caudate nucleus, DLPFC, and hippocampus. **A.** Circos plot showing significant differentially expressed genes (DEGs) for the caudate nucleus (blue; n=393; 121 female and 272 male), DLPFC (red; n=359; 114 female and 245 male), and hippocampus (green; n=375; 121 female and 254 male) across all chromosomes. Female bias (upregulated in female individuals) in red, and male bias (upregulated in male individuals) in blue. **B.** Gene set enrichment analysis (GSEA) of sex differential expression analysis across brain regions, highlighting terms upregulated in females (female bias) or males (male bias). NES: normalized enrichment score. XCI: X-chromosome inactivation. **C.** Scatterplots of the estimated proportion of expression variance explained by sex within the 100 most significant autosomal DEGs (i.e., adjusted p-value) for the caudate nucleus (DEGs, n=60), DLPFC (DEGs, n=25), and hippocampus (DEGs, n=42). **D.** UpSet plot showing overlap of DEGs across the caudate nucleus, DLPFC, and hippocampus. Blue is shared across the caudate nucleus, DLPFC, and hippocampus; orange, shared between two brain regions; and black, unique to a specific brain region. * Indicating p-value < 0.0001 for two-sided, Fisher's exact test. **E.** Scatterplots of effect size (logFC) for all genes tested showing concordant positive directionality with significant two-sided, Spearman correlation (R^2) of all genes. A fitted trend line is presented in blue as the mean values +/- standard deviation. **F.** Example box plots of genes showing an interaction between sex and brain region. FC= fold change log₂ (male / female). Female individuals in red and male individuals in blue. Adjusted

p-value (P) annotation using `dream`⁵ (default of Satterthwaite approximation) generated statistics annotation. Box plots show the median and first and third quartiles, and whiskers extend to $1.5\times$ the interquartile range.

Fig. 3. Transcriptional changes for schizophrenia shared between sexes within brain regions. RRHO (rank-rank hypergeometric overlap) maps comparing schizophrenia transcriptional changes for all genes between brain region pairs stratified by direction of effect in **A.** females and **B.** males. The panel presents the overlapping relationship between two brain regions. The color bars represent the degree of significance [$-\log_{10}(\text{p-value})$] of overlap between two brain regions. Arrows show the direction of effect for schizophrenia (upregulated or downregulated in schizophrenia) by brain region. **C.** RRHO map comparing female and male schizophrenia transcriptional changes within brain regions for all genes. The color bar represents the degree of significance [$-\log_{10}(\text{p-value})$] of the overlap between the sexes. **D.** Venn diagram showing overlap within brain regions for sex-stratified schizophrenia differentially expressed genes (DEGs; female in red and male in blue; $\text{FDR} < 0.05$). Female-specific schizophrenia DEGs in red, male-specific schizophrenia DEGs in blue, and schizophrenia DEGs shared between female- and male-specific schizophrenia analyses in purple. SZ: schizophrenia.

Fig. 4. Sex-interacting eQTL (si-eQTL) are shared across brain regions. **A.** UpSet plot displaying overlap across brain regions for si-eQTL (local false sign rate [lfsr] < 0.05). Blue is shared across the caudate nucleus (n=399; 126 female and 273 male), DLPFC (n=377; 121 female and 256 male), and hippocampus (n=394; 126 female and 268 male); orange, shared between two brain regions; and black, unique to a specific brain region. **B.** Heatmap of the proportion of gene level si-eQTL sharing with sign match (left), within a factor of 0.5 effect size (middle), and within a factor 0.99 effect size (right). **C.** Functional enrichment plot of the ten most significant gene ontology-terms (biological processes) of eGenes for the caudate nucleus (blue), DLPFC (yellow), and hippocampus (gray).

Fig. S6. Autosomal sex-specific DEGs show significant correlation with sex and strong predictive power for sex in the brain. Scatterplots of principal components (PC) 1 and 2 from dimensionally reduced expression of all allosomal and autosomal DEGs for the caudate nucleus, DLPFC, and hippocampus.

Fig. S10. Significant sharing of sex-specific differentially expressed genes (DEGs) across brain regions replicate in the CommonMind Consortium (CMC) DLPFC. UpSet plot showing number of DEGs shared across brain regions with the CMC DLPFC cohort **A.** NIMH HBCC and **B.** MSSM-Penn-Pitt. * Indicating p-value < 0.01 for two-tailed, Fisher's exact test. Green is shared across the four brain regions; blue, shared across three brain regions; orange, shared between two brain regions; and black, unique to a specific brain region.

Fig. S12. Concordant transcriptional changes for sex-specific expression across the brain. RRHO (rank-rank hypergeometric overlap) maps comparing sex-specific transcriptional changes for all genes between brain region pairs stratified by direction of effect. There are no genes with discordant direction of effect. The panel presents the overlapping relationship between two brain regions. The color bar represents the degree of significance [$-\log_{10}(\text{p-value})$] of overlap between two brain regions. Arrows show the direction of effect for sex (female biased [down-regulated] or male biased [up-regulated]) by brain region.

A

$$\text{Relative X Expression (RXE)} = \log_2(X) - \log_2(A)$$

**B**
Fig. S18. Replication of relative X expression (RXE) sex differences within the DLPFC of CommonMind Consortium (CMC). **A.** Schematic of RXE (left) and box plots showing RXE comparison between female (red) and male (blue) individuals for the CMC DLPFC (right). **B.** Box plots showing RXE comparison between neurotypical controls (gray) and schizophrenia (gold) individuals for female (left) and male (right) individuals in the CMC DLPFC. Box plots show the median and first and third quartiles, and whiskers extend to 1.5× the interquartile range. Annotations are p-values from Mann-Whitney U two-tailed test.

Fig. S27. si-eQTLs are shared across brain regions. UpSet plots showing the majority of si-eQTL are shared across features for eTranscripts (si-eQTL associated with unique transcripts), eExons (si-eQTL associated with unique exons), and eJunctions (si-eQTL associated with unique junctions). Blue is shared across three brain regions; orange, shared between two brain regions; and black, unique to a specific brain region.

$$RXE = \log_2(\text{mean TPM of X-chromosome genes}) - \log_2(\text{mean TPM of all autosomal genes})$$

Equation 9

References

1. McCarthy, M. M. Stress during pregnancy: Fetal males pay the price. *Proc Natl Acad Sci USA* **116**, 23877–23879 (2019).
2. Benjamin, K. J. M. *et al.* Analysis of the caudate nucleus transcriptome in individuals with schizophrenia highlights effects of antipsychotics and new risk genes. *Nat. Neurosci.* **25**, 1559–1568 (2022).
3. Collado-Torres, L. *et al.* Regional Heterogeneity in Gene Expression, Regulation, and Coherence in the Frontal Cortex and Hippocampus across Development and Schizophrenia. *Neuron* **103**, 203-216.e8 (2019).
4. Hoffman, G. E. *et al.* Sex differences in the human brain transcriptome of cases with schizophrenia. *Biol. Psychiatry* **91**, 92–101 (2022).
5. Hoffman, G. E. & Roussos, P. Dream: powerful differential expression analysis for repeated measures designs. *Bioinformatics* **37**, 192–201 (2021).

REVIEWERS' COMMENTS

Reviewer #1 (Remarks to the Author):

The revised manuscript addressed all review comments thoroughly and concisely. One minor point that could be addressed without need for any additional review, are the sex-biased genes on X and Y: male-bias of PAR1 genes is not surprising (pg. 19), as it has been previously reported across a number of species:

<https://pubmed.ncbi.nlm.nih.gov/26719789/>

This likely reflects residual dampening of PAR1 on the inactive X. As the authors know, escapee genes outside of PAR1 are female-biased given their increased copy relative to males. Although not a major point of the study, the authors may want to consider clarifying this point in Fig. S17, for example choosing a heatmap representation over the current venn diagrams.

Some additional detail about the reduced RXE in the male hippocampus would be helpful as well, so the reader doesn't have to refer to ref.52: namely, to state whether reduced RXE stems from genes subject to XCI in females, or either of the escapee categories (male-biased PAR1, vs. female-biased)?

Finally, the RPS10P3 finding is interesting, as it falls into a relatively gene-poor region but near potential long-range enhancers. This raises the question whether expression of any of the neighboring genes on chr9 may mediate sex-specific effects in any of the brain regions, or have been previously associated with schizophrenia. Something to potentially add in the discussion if there were any references to this point.

Reviewer #3 (Remarks to the Author):

Authors did an extensive work with reanalyzing their data. All my concerns have been addressed.

We thank the reviewers for the comments and suggestions. We have made the additional suggestions as described below. We believe this improves the manuscript clarity.

Reviewer #1 (Remarks to the Author):

Reviewer comment: The revised manuscript addressed all review comments thoroughly and concisely. One minor point that could be addressed without need for any additional review, are the sex-biased genes on X and Y: male-bias of PAR1 genes is not surprising (pg. 19), as it has been previously reported across a number of species: <https://pubmed.ncbi.nlm.nih.gov/26719789/>

Benjamin response: We have updated the main text and discussion to reflect this information.

Main text excerpt: Across all three brain regions, we found XCI escape genes were significantly enriched within the female- and male-biased DEGs (Fisher's exact test, Bonferroni < 0.01; **Fig. 2A**). Moreover, all male-biased escaping XCI genes were located on the PAR (pseudoautosomal regions) of both X and Y chromosomes (*AKAP17A*, *ASMTL*, *ASMTL-AS1*, *CD99*, *CD99P1*, *DHRXS*, *GTPBP6*, *IL3RA*, *LINC00106*, *PLCXD1*, *PPP2R3B*, and *ZBED1*; **Data S9**). This finding aligns with previous reports showing a male-biased enrichment of escaping XCI genes on PAR1^{36,37}.

Discussion excerpt: Here, our analysis aligned with previous work showing an enrichment of genes known to escape XCI^{34,35}, including male-bias enrichment for PAR genes^{36,37}.

Reviewer comment: This likely reflects residual dampening of PAR1 on the inactive X. As the authors know, escapee genes outside of PAR1 are female-biased given their increased copy relative to males. Although not a major point of the study, the authors may want to consider clarifying this point in Fig. S17, for example choosing a heatmap representation over the current venn diagrams.

Benjamin response: To improve clarity, we have included heatmap plots (**Fig. S17**) as suggested by the reviewer.

Reviewer comment: Some additional detail about the reduced RXE in the male hippocampus would be helpful as well, so the reader doesn't have to refer to ref.52: namely, to state whether reduced RXE stems from genes subject to XCI in females, or either of the escapee categories (male-biased PAR1, vs. female-biased)?

Benjamin response: We appreciate the reviewer's suggestion for further clarification. We have added information about the source of the reduced RXE in the male hippocampus. Specifically, the significant decrease in RXE stems primarily from reduced expression of the inactive XCI genes in male patients with schizophrenia (**Fig. S21**).

Reviewer comment: Finally, the *RPS10P3* finding is interesting, as it falls into a relatively gene-poor region but near potential long-range enhancers. This raises the question whether expression of any of the neighboring genes on chr9 may mediate sex-specific effects in any of the brain regions, or have been previously associated with schizophrenia. Something to potentially add in the discussion if there were any references to this point.

Benjamin response: We appreciate the suggestion regarding *RPS10P3*. While there is no previously associated schizophrenia association with *RPS10P3*, we updated the discussion to acknowledge the potential association with long-range enhancers, expanding on its previously known links to diverse traits (refs 25-29). This strengthens our findings by suggesting a potential mechanism underlying its role in sex-specific prediction across the brain.

Excerpt: Notably, the autosomal pseudogene *RPS10P3* emerged as a key driver of sex prediction across the brain. Located in a gene-poor region, *RPS10P3* is flanked by enhancers and has prior links to diverse traits²⁵⁻²⁹, including sex-interacting cleft lip²⁹ and psychosis-related lateral ventricle temporal horn volume²⁸.

Reviewer #3 (Remarks to the Author):

Reviewer comment: Authors did an extensive work with reanalyzing their data. All my concerns have been addressed.

Benjamin response: We are pleased to confirm that we have addressed all the reviewer's concerns.

Fig. S17. Majority of XCI (X-chromosome inactivation) escaping genes are upregulated in female individuals across the brain. Heatmap plots of residualized expression for XCI differentially expressed genes for the caudate nucleus (n=393; 121 female and 272 male), dorsolateral prefrontal cortex (DLPFC; n=359; 114 female and 245 male), and hippocampus (n=375; 121 female and 254 male). Heatmap rows are annotated for sex (female: red, male: blue) and columns are annotated for XCI status (escape: blue, variable: orange, inactive: green).

Fig. S21. Significant decrease in relative X expression (RXE) in the hippocampus of male patients with schizophrenia driven by a reduction of inactive X-chromosome inactivation (XCI) genes. Box plots of RXE expression between neurotypical controls (gray; n=188) and schizophrenia (gold; n=88) individuals in the hippocampus of male individuals (n=276). RXE expression calculated using all genes, no escaping XCI genes, no variable XCI genes, and no inactive XCI genes. Box plots show the median and first and third quartiles, and whiskers extend to 1.5× the interquartile range. Annotations are p-values from Mann-Whitney U two-tailed test.